# Beyond Soil-Dwelling Actinobacteria: Fantastic Antibiotics and Where to Find Them

**DOI:** 10.3390/antibiotics11020195

**Published:** 2022-02-02

**Authors:** Javier Santos-Aberturas, Natalia M. Vior

**Affiliations:** Department of Molecular Microbiology, John Innes Centre, Norwich NR7 4UH, UK

**Keywords:** antibiotics, secondary metabolites, bacterial diversity, genome mining, symbiosis, unculturable bacteria, entomopathogenic bacteria

## Abstract

Bacterial secondary metabolites represent an invaluable source of bioactive molecules for the pharmaceutical and agrochemical industries. Although screening campaigns for the discovery of new compounds have traditionally been strongly biased towards the study of soil-dwelling Actinobacteria, the current antibiotic resistance and discovery crisis has brought a considerable amount of attention to the study of previously neglected bacterial sources of secondary metabolites. The development and application of new screening, sequencing, genetic manipulation, cultivation and bioinformatic techniques have revealed several other groups of bacteria as producers of striking chemical novelty. Biosynthetic machineries evolved from independent taxonomic origins and under completely different ecological requirements and selective pressures are responsible for these structural innovations. In this review, we summarize the most important discoveries related to secondary metabolites from alternative bacterial sources, trying to provide the reader with a broad perspective on how technical novelties have facilitated the access to the bacterial metabolic dark matter.

## 1. Introduction

Bacterial specialized metabolites (also known as secondary metabolites or natural products) constitute one of the most astonishing expressions of biodiversity, as well as a major source of bioactive molecules for the pharmaceutical and agrochemical industries. Directly or indirectly linked to their ecological roles in nature, these molecules exhibit a wide range of biological activities that have fundamentally changed medical and agronomical practice, including antibiotics (antibacterial, antifungal and anticancer compounds), antiparasitic drugs, insecticides and immunosuppressants, among many others. Specialized metabolites are generally complex and structurally unusual molecules but despite this, bacteria synthesize them employing just ordinary primary metabolites as building blocks. Such transformations are performed by highly specialized enzymes that in many cases catalyse remarkably exotic chemical reactions, making them a central focus of interest for the development of biocatalysts in green chemistry. The enzymes involved in the biosynthesis of a given natural product are generally encoded by genes compactly grouped within the bacterial genomes, conforming functional units known as biosynthetic genes clusters (BGCs).

Traditionally, and since Selman A. Waksman’s germinal studies that led to the discovery of actinomycin and streptomycin, soil-dwelling Actinobacteria (and most prominently those belonging to *Streptomyces*) have constituted the main source of clinically relevant bacterial natural products, both for fundamental and practical reasons: *Streptomyces* and related genera treasure an extraordinary biosynthetic potential, most of it still unveiled, as revealed since the entrance of these groups of bacteria into the genomic era [1,2,3]; they inhabit complex and changing environments involving multiple interactions with other organisms where specialized metabolites play a central role; and, unlike most bacteria [4,5], they are easily isolated and cultured under laboratory conditions. Thus, most bacterial natural products screening campaigns have been strongly biased towards soil-dwelling Actinobacteria during the last eight decades. However, in recent years, the antibiotic resistance and discovery crisis [6,7] has triggered radical innovations in the natural products discovery strategies in the search for compounds able to fight the life-threatening infections caused by multidrug-resistant pathogens (especially those belonging to the ESKAPE group [8]), as well as to satisfy previously unmet therapeutic needs. Some of those strategies are focused on the genetic engineering-mediated awakening of silent or cryptic BGCs previously detected by the bioinformatic analysis of bacterial genomes. This approach is known as “genome mining”, and its philosophy, achievements and limitations have been extensively reviewed [9,10,11].

At the same time, the exploration of new phylogenetic groups of bacteria and new ecological niches is becoming a powerful trend in the natural product discovery field. Since the advent of affordable whole genome sequencing, it has become clear that many groups of previously neglected bacteria treasure a vast potential to produce specialized metabolites. Supported by the development of new cultivation [12,13,14,15], sequencing [16,17], gene cloning [18], bioinformatic [19,20,21,22,23,24] and screening techniques [25], these strategies are facilitating the access to a very substantial portion of the so-called “metabolic dark matter” never explored before. This includes completely novel chemical diversity evolved from independent phylogenetic origins and under the demands of different selective pressures, biological interactions and molecular targets. In this review, we summarize the most important achievements in the exploration of new bacterial biodiversity and ecology for the discovery of novel specialized metabolites.

## 2. Bacterial Symbionts of Marine Invertebrates

Except for notable exceptions [26], animals lack the metabolic diversity exhibited by microbes and plants, but they have developed symbiotic relationships with bacteria multiple times through evolution, thus gaining access to the ecological advantages of the bacterial metabolic repertoire. In some cases, the nature of the symbiosis is linked to primary metabolic needs (i.e., in the case of corals dependent on their photosynthetic symbionts), but in many situations, especially in animals lacking other protection systems, the need for chemical defence against UV radiation, infections or predators has constituted the selective pressure driving symbiosis establishment [27]. Marine invertebrates (especially sessile ones) are an excellent example of this. With their extraordinary diversity (at least 175,000 extant species belonging to almost all animal phyla) and with little evolutionary overlap with terrestrial soil ecosystems, marine invertebrates constitute a promising source of unprecedented chemical structures and novel biological activities. Although in most cases bacterial symbionts have been presumed to be the source of the specialized metabolites found in marine invertebrates, the experimental evidence has often been elusive to traditional approaches, and still represents a major technical challenge. The high complexity of the microbial communities associated with some marine invertebrates (e.g., sponges) and the frequent difficulties culturing their bacterial symbionts using standard microbiology techniques have hampered for many years the progress in the knowledge about their biosynthetic pathways. Only recently, the application of culture-independent approaches, as metagenomics, cell sorting, and single-cell sequencing has boosted this area of research [28]. These developments will be crucial not only from a basic knowledge perspective, but also for the commercial supply of marine invertebrate-derived molecules, as their natural sources are often scarce and slow-growing and their harvest harmful to natural environments. Thus, the industrial production of these compounds will most likely rely on the heterologous expression of their biosynthetic pathways and/or on the application of synthetic or semisynthetic approaches [28]. In the following subsections we will dissect some of the most recent and interesting examples of specialized metabolite discovery from different groups of marine invertebrates.

### 2.1. Sponges

With almost 9000 living species, sponges (phylum Porifera) represent the most ancient group of Metazoans. Sponges are sessile organisms and lack true tissues or organs, feeding by filtering food particles (mainly bacteria and suspended organic matter) out of the water flowing though their porous bodies. Unlike other groups of animals, sponges are unable to produce functional mucus layers, and instead have their surfaces covered by a layer of microbial symbionts that can represent up to 40–60% of the sponge’s wet weight and plays multiple roles in the life of the host, including nitrogen fixation, waste processing and nutrient transfer [29,30]. These symbionts are also believed to be responsible for the biosynthesis of the great diversity of secondary metabolites found in sponges, as pointed out by germinal studies [31], but most of the time they have shown to be unculturable using standard microbiological techniques. Culture-independent approaches have thus been crucial for the understanding of the sponge bacterial symbionts. 16 S rRNA metagenomic analyses has shown that, while some bacterial symbionts seem to be almost ubiquitous across unrelated sponge hosts, others seem to be highly specialized. Interestingly, many of the symbionts identified by metagenomic analysis show no obvious phylogenetic relationship with culturable groups of bacteria, as in the case of the new candidate phyla Poribacteria [32] and Tectomicrobia [33].

Polyketides, a major class of natural products including multiple medicinal representatives, constitute an excellent example of the innovation found among the sponge symbionts’ biosynthetic capabilities. Polyketide biosynthesis is achieved by the decarboxylative condensation of short acyl-CoA building blocks by enzymatic systems called Polyketide Synthases (PKSs), in an analogous manner to fatty acid biosynthesis [34]. Type I PKSs are giant multifunctional enzymes organized in modules, each module including several enzymatic domains responsible for the selection of a building block, its condensation into the growing polyketide chain and its modification by a variable degree of reductive steps. As type I PKSs are exclusively found in Bacteria, many clinically interesting sponge-derived polyketides resembling type I PKS products, such as the potential antitumor agents peluroside A, laulimalide, swinholides, misakinolides or onnamides (Figure 1), have been traditionally assumed to be of bacterial origin. In the case of the onnamides (from the sponge *Theonella swinhoei*, reported as the source of a least 40 different bioactive compounds), this origin has been formally proved. Beyond these particular examples, the potential of sponges as sources of new polyketides seems enormous, as shown by the metagenomic analysis of the symbiotic communities associated with multiple sponges [35]. In this study, the cloning and sequencing of PKS ketosynthase (KS) domains revealed that only 8% of them could be assigned to previously known PKS families.

One of the most important examples illustrating the metabolic potential of sponge symbionts is the characterization of polytheonamides (Figure 1), cytotoxic peptides of high structural complexity originally isolated from the sponge *Theonella swinhoei* [36], which constituted the first representatives of the proteusin family of peptides. Due to the presence of multiple non-proteinogenic amino acids within their structures, these compounds were originally suspected to be the product of a Non-Ribosomal Peptide Synthetase (NRPS) assembly line. NRPSs are modular enzymatic systems that use amino acids as building blocks to make peptides in an analogous manner to the way PKSs use acyl-CoA to build polyketides [37]. However, the metagenomic analysis of the bacterial symbiotic community associated with the sponge revealed that polytheonamides are Ribosomally synthetized and Post-translationally modified Peptides (RiPPs) [38,39], and therefore derived from genetically encoded peptides. Differential centrifugation, fluorescence-assisted cell sorting (FACS) and single-cell genome sequencing resulted in the identification of a single filamentous bacterial symbiont (“*Candidatus* Entotheonella sp.”, belonging to the phylum “*Candidatus* Tectomicrobia”) as the producer of almost all known bioactive peptides and polyketides from *Theonella swinhoei*. This includes polytheonamides and onnamides, as well as several NRPS products, like nazumamide A, cyclotheonamide and keramamide. So far unculturable, “Entotheonella” species carry large genomes (ca. 9 MB), seem to be ubiquitous among sponges and exhibit an extraordinary potential as sources of novel bioactive specialized metabolites. However, less exotic groups of symbionts are also important sources of sponge holobiont natural products. For example, endosymbiotic cyanobacteria have been identified as the producers of the highly toxic polybrominated diphenyl ethers found in sponges of the family Dysideidae [40]. The enormous amount of information generated by metagenomic sequencing of these complex microbial communities constitutes excellent database material for genome mining, as recently demonstrated by the discovery of a widespread family of brominated RiPPs [41]. This implies that the metabolic potential of these symbiotic communities could also include numerous cryptic compounds.

### 2.2. Tunicates

Closely related to vertebrates, tunicates (phylum Chordata, subphylum Tunicata) comprise 3000 species of mainly sessile, water-filtering animals. Tunicates are a rich source of natural products, with more than 1200 compounds isolated from them so far, including polyketides, alkaloids and peptides [42,43]. However, the microbial communities hosted by Tunicates are not as complex as in the case of sponges, being dominated by the presence of Cyanobacteria, Proteobacteria and Actinomycetales [42]. From the nine marine-derived drugs approved for therapeutical use up to 2020, two are derived from tunicates: trabectedin (also known as ET-743 or Yondelis^®^) and plitidepsin (also known as dehydrodidemnin B or Aplidin^®^), both used as anticancer drugs.

Trabectedin (Figure 2) is a tetrahydroisoquinoline alkaloid compound originally isolated from the tunicate *Ecteinascidia turbinata*. The structural similarities of the core of these molecules with other bacterial natural products suggested from the beginning a symbiotic origin for this compound: in fact, the industrial production of Yondelis^®^ by Pharmamar is currently achieved by a multi-step semisynthetic approach using as starting material cyosafracin B, a tetrahydroisoquinoline compound produced by the fermentation of *Pseudomonas fluorescens* [44]. The bacterial origin of trabectedin was confirmed by the identification of its BGC by a metagenomic analysis searching for key biosynthetic genes [45]. The γ-proteobacterium “*Candidatus* Endoecteinascidia frumentensis” has been identified as the trabectedin producer. This organism is an obligate endosymbiont with an intensively reduced genome (0.6 MB), indicative of a long co-evolution process alongside its tunicate host [46]. Lurbenectidin (Zepzelca^®^), a trabectedin synthetic derivative also produced by Pharmamar, has recently received FDA-accelerated approval for the treatment of small cell lung cancer [47].

Apart from its antineoplastic properties, plitidepsin (Figure 2) has shown an important potential as an antiviral drug, including a potent preclinical activity against COVID-19 [48]. This NRPS-PKS hybrid depsipeptide was isolated from the tunicate *Aplidium albicans*. A closely related congener of plitidepsin, didemnin B, was isolated from another tunicate (*Trididemnum* sp.). Although the precise symbiont producing didemnins in these tunicates has not been identified yet, their bacterial origin is strongly suspected, as two species of free-living marine α-proteobacteria, *Tristella mobilis* and *Tristella bauzaunensis*, have been reported as producers of these compounds [49,50].

Patellamides (Figure 2), a group of cytotoxic cyclic peptides from *Lissoclinum patella*, constitute another interesting example of the biosynthetic potential of tunicate bacterial symbionts, in this case the cyanobacterial endosymbiont *Prochloron didemni*. Despite being thought to be the product of a NRPS assembly line, the genome sequence of the purified endosymbiont did not show NRPS-containing BGCs that could justify the biosynthesis of patellamides. However, the identification of a genetically encoded peptide matching the patellamides amino acid sequences revealed that these compounds are RiPPs [51,52]. Subsequently, many patellamide and related RiPP BGCs have been identified and their products isolated from multiple free-living cyanobacteria, as will be discussed later in this review. *P. didemni* has also been shown to produce a lanthipeptide with anti-HIV activity, divamide A [53]. Lanthipeptides are a family of RiPPs characterized by the presence of lanthionine bonds within their structures and, although well represented across Actinomycetes, were previously considered rare in marine animals. Unlike other endosymbionts and despite decades of failed cultivation attempts, *P. didemni* genome does not show signs of reduction and apparently remains fully metabolically functional [54]. The symbiotic community associated with *Lissoclinum patella* includes other uncultivated bacteria that also seem to treasure potential for the production of additional secondary metabolites [54], as in the case of the patellazoles (Figure 2), nanomolar concentration inhibitors of the protein synthesis [55]. Patellazoles are the product of a trans-AT PKS system [56] encoded within the notably reduced genome (1.48 Mb) of a symbiotic α-proteobacterium, “*Candidatus* Endolissoclinum faulkneri” [57].

### 2.3. Other Groups of Marine Invertebrates

Although sponges and tunicates are probably the most extensively studied sources of specialized metabolites from marine invertebrates, it cannot be ignored that in fact these two groups of animals represent just a small fraction of the invertebrate diversity inhabiting the sea. Many other groups, such as Bryozoans (phylum Bryozoa) [58], shipworms (phylum Mollusca, class Bivalvia, order Myida, family Teredinidae) [59] or bristle worms (phylum Annelida, class Polychaeta) [60], among many others, have been identified as interesting sources of new molecules.

Bryostatins [61] were originally isolated from *Bugula neritina* and represent the best- studied group of metabolites produced by the 6000 existing species of Bryozoans. These compounds are trans-AT PKS products and their role in nature is the protection of the larvae of the bryozoan from predators [62]. Bryostatins (Figure 3) act as protein kinase C modulators, a biological activity with great potential for biomedical applications. The unculturable symbiotic γ-proteobacteria “*Candidatus* Endobugula sertula” was identified as the bryostatin producer, based on molecular and microscopic techniques [63,64,65,66].

Shipworms are essential for the recycling of wood in the sea, and lately have been revealed as an interesting source of bioactive compounds. In contrast to most of the cases exposed until now, *Teredinibacter turnerae*, the γ-proteobacterial endosymbiont responsible for the biosynthesis of the natural products isolated in several species of shipworms (as well as for the supply of the lignocellulosic enzymes required to sustain their host lifestyle) is culturable, a fact that has enormously facilitated its study. *T. turnerae* natural products show remarkable biological activities, as in the case of the turnercyclamycins (Figure 3), cyclic lipopeptides displaying biological activity against the life-threatening ESKAPE pathogen *Acinetobacter baumannii* [67]. *T. turnerae* is also able to produce boronated polyketides belonging to the tartrolon class [68,69], with tartrolon E displaying a antibiotic activity in the nanomolar range against a wide diversity of apicomplexan parasites [70]. The genome of *T. turnerae* (5.1 Mb) shows no reduction process, is highly enriched in complex polysaccharide metabolism enzymes and carries a total of nine BGCs containing NRPS or PKS enzymes [71], thus promising additional biosynthetic potential.

Finally, another relevant set of molecules originally associated with marine invertebrates are dolastatins (Figure 3), potent cytotoxic molecules first isolated in 1987 from the sea hare *Dolabella auricularia* (phylum Mollusca, class Gastropoda) [72]. These molecules, in particular dolastatin 10 and 15, are strong cytotoxic compounds that depolymerize microtubules, causing cell-cycle arrest [73]. They have high clinical relevance but given their toxicity, current treatments with synthetic derivatives of these molecules, such as monomethyl auristatin E, are carried out in the form of antibody conjugates (e.g., Brentuximab vedotin, Adcetris^®^, for the treatment of Hodgkin’s lymphoma), to reduce their off-target effects [74]. In the past few years, multiple dolastatins and analogues have been discovered in free-living cyanobacteria, leading to the conclusion that the dolastatin in sea hare likely has dietary origin [75,76,77].

## 3. Insect-Associated Bacteria

With one million described species, Insects (phylum Artropoda, subphylum Hexapoda, class Insecta) are the most diverse and abundant group of animals. Insects can be found in all the habitats on Earth, although only a few species inhabit the oceans, and they have evolved to exploit almost every imaginable food resource. Thus, their lifetsyles and ecological roles truly challenge the limits of imagination [78]. Therefore, it is easy to assume that they might have established all sorts of interactions with microorganisms, and that in many cases those interactions are based on the adaptative advantages offered by bacterial secondary metabolism.

Ants (order Hymenoptera, family Formicidae) establish mutualistic associations that very likely constitute the best studied model to understand the importance of insect–bacteria interactions in nature. Fungus-growing ants employ different plant sources to farm a fungus which constitutes their single source of food. To protect their fungal garden from specialized fungal parasites (like *Escovopsis* spp.) the ants use bacterial symbionts (typically Actinobacteria belonging to *Pseudonocardia* [79,80], *Streptomyces* [81,82] or *Saccharopolyspora* [83], but also β-proteobacterial symbionts like *Burkholderia* [84]) that produce antifungal compounds that inhibit the invasion by the parasitic fungus [85,86]. Since the identification of the first of those antifungals, the depsipeptide dentigerumycin [87] (Figure 4), many other novel compounds have been isolated from bacteria associated with fungus-growing ants, like the halogenated NRPS-PKS hybrids colibrimycins [88] and collismycins [89] (Figure 4), novel quinones [90], diverse macrolides showing both antifungal and antileishmanial activities [91] or the non-ribosomal peptide attinimicin, an iron-dependent antifungal [92]. Formicamycins (Figure 4), pentacyclic type II PKS products from *Streptomyces formicae* (a strain isolated from the tree-associated *Tetraponera* ants which shows an impressive secondary metabolism potential [93]), show potent activity against Gram-positive pathogens without apparent generation of spontaneous resistance [94,95,96,97]. Formicamycins have also been recently isolated from a soil *Streptomyces* strain [98]. The degree of specificity and co-evolution in the ant-actinobacteria associations has been a matter of debate [80,99,100], although ants from the Attini tribe (leaf-cutter ants) have evolved specific structures to host *Pseudonocardia* spp. in their cuticles [101], suggesting that at least in some cases the associations have co-evolutionary implications linked to relevant ecological roles. Other Hymenoptera have also established symbiotic relationships with Actinobacteria, as in the case of the solitary bee wolves, whose pupae and eggs are protected from microbial infections by the antibiotics produced by specific bacteria inhabiting specialized structures in the antennae of the adults [102,103,104].

Despite the ecological beauty of ant mutualistic models, the production of pederin (Figure 4) antibiotics by *Paederus* beetles (order Coleoptera, family Staphylinidae) arguably constitutes a much more striking example of how the exploration of new biological sources can lead to the discovery of completely new biochemistry. Pederins are highly active anticancer polyketides closely related to the sponge-related onnamides (in fact, the previous knowledge about the pederin BGC was key for the identification of the onnamides as bacterial secondary metabolites), and they are produced for chemical defence against predators. PCR screening of a metagenomic library built using total DNA from *Paederus* beetles in the search for specific PKS domains (in particular, KS domains) led to the discovery of the PKS BGC responsible for the production of these compounds. Unexpectedly, none of the type I PKS modules present in the assembly line included an acyltransferase (AT) domain, but two discrete AT-domain proteins were encoded within the pederin BGC instead [105]. In addition to this, unprecedented catalytic domains were found within the pederin PKS modules, remarkably expanding the possibilities of the known polyketide biochemistry [106]. This BGC represented the first example of a trans-AT PKS system, a vast group of PKS assembly lines with an independent evolutionary origin from standard (cis-AT) type I PKSs [56]. Because of their rarity among Actinobacteria trans-AT PKSs had been overlooked before, but the genomic analysis of different groups of bacteria indicates that in fact they are widespread in nature, accounting for almost 40% of all the bacterial PKSs. An uncultivated endosymbiont closely related to *Pseudomonas aeruginosa* has been identified as the true producer of pederin in *Paederus* beetles [107]. In the Asian citrus psyllid (order Hemiptera, family Liviidae) a pederin analog is produced by a β-proteobacterial endosymbiont with a highly reduced genome (“*Candidatus* Profftella armatura”) [108]. Other pederin analogs have been recently isolated from free-living α-proteobacteria (*Labrenzia* sp.) [109] and Cyanobacteria [110], thus suggesting that this family of compounds is probably widespread in nature.

As a last example of the wealth of specialized metabolites derived from insect symbionts, we would like to mention the production of lagriamides (Figure 4), trans-AT-PKS-NRPS hybrid antifungal compounds which protect the offspring of the beetle *Lagria villosa* against pathogens [111]. The eggs of this beetle harbour multiple symbionts related to *Burkholderia gladioli*, but only one of them is responsible for the production of the lagriamides. Interestingly, this symbiont is the only member of the egg microbiome that shows genome reduction, a process that most likely started after acquiring the lagriamides’ BGC by horizontal gene transfer [112]. *Burkholderia* species represent a very interesting source of new natural products, and not only in association with insects, as we will discuss below.

## 4. Entomopathogenic Bacteria

Entomopathogenic bacteria have been long harnessed for their potential as biocontrol agents in agriculture, complementing or even substituting synthetic pesticide use, which often presents complications due to off-target toxicity and resistance generation [113]. So far, over a hundred bacterial species have been characterized as arthropod pathogens, with Gram-positives belonging to the genus *Bacillus* among the most well-studied and widely used as “microbial insecticides” [114,115]. Other bacterial families with recognized insecticidal potential are Paenibacillae, as well as the Gram-negative Enterobacteriaceae and Neisseriaceae. Pathogenicity in these bacteria relies mainly on the production of an assortment of degradation enzymes, including proteases, chitinases and other toxins that lead to mid-gut toxicity and death [116,117]. On closer inspection, and thanks to the expansion of the next generation sequencing techniques, it has become apparent that many of these bacteria actually have the potential to produce a much wider array of molecules with diverse functions, including many secondary metabolites.

Among the known entomopathogenic bacteria, the best studied representatives with regards to their secondary metabolite production potential are without a doubt the enterobacteria belonging to the genera *Photorhabdus* and *Xenorhabdus*. These bacteria have a complex life cycle, in which they enter a mutualistic relationship with entomopathogenic nematodes of the genera *Steinernema* and *Heterorhabditis*, respectively. These nematodes, while carrying the bacteria in their gut, infect soil-dwelling insect larvae and once inside release the bacteria into their haemocoel, where they start producing multiple proteins and secondary metabolites that inhibit the insect immune response, supress competitor microorganism growth and allow both bacterial and nematode multiplication, leading to larval death. Just within the genus *Xenorhabdus* more than 23 secondary metabolite families [118] have been identified so far and a recent analysis of the BGC distribution in a panel of dozens of strains from the *Photorhabdus*/*Xenorhabdus* clade showed that each of them contains between 21 and 41 BCGs, many of them still uncharacterized, highlighting their enormous biosynthetic potential [119]. For comprehensive overviews of the secondary metabolites produced by these two genera and their bioactivities, please refer to genomic studies and reviews focused on these organisms [118,119,120,121]. The majority of the BGCs identified so far in these organisms encode NRPS or hybrid NRPS-PKS as well as some RiPPs, so in consequence most of the natural products isolated from these bacteria are peptide or peptide-derived in nature [118,119,120]. (Figure 5). There are, however, many other structurally diverse metabolites that assist these bacteria in maintaining their particular lifestyle. The isonitrile insecticide rhabduscin (Figure 5), nearly ubiquitous in both *Photorhabdus* and *Xenorhabdus* strains, acts as inhibitor of the insect phenoloxidase enzyme, essential for its immune response [122]. Another immunosuppressant molecule is benzylideneacetone (Figure 5) which inhibits phospholipase A(2), preventing biosynthesis of immune-mediating eicosanoids [123,124]. Isopropylstilbenes (Figure 5), mainly found in *Photorhabdus*, exhibit multiple activities, including antimicrobial, antifungal, inmunomodulator as well as stimulating symbiosis establishment with the nematode host [120,125].

Diverse *Xenorhabdus* strains have been reported to produce numerous NRPS-derived molecules: xenematides, xenobactin, szentiamide, bicornutin, nematophin, PAX peptides, gameX peptides, xenortides and rhabdopeptides, among others. Xenematides (Figure 5), xenobactin and szentiamide are all cyclic depsipeptide antibiotics first isolated from *X. nematophila*, *Xenorhabditis* sp. PB30.3 and *X. szentirmaii* DSM 16338, respectively [126,127,128]. Besides their antimicrobial activity, some xenematides are weak insecticidals [126] and szentiamide has been reported to have both insecticidal and antiprotozoal activities [129], whereas xenobactin exhibits antiprotozal activity [127]. The xenematide repertoire has recently been expanded with two novel molecules (F and G) thanks to a targeted PCR screening of a *Xenorhabdus* strain collection looking for depsipeptide biosynthesis genes [130]. Untargeted metabolomic screening has also revealed novel depsipeptide families such as xentrivalpeptides, with up to 17 congeners but no known biological function [131] or fatflabets and xeneprotides (Figure 5), identified first through mass spectral metabolic networking of culture extracts that were then crosschecked with the genomic information of the producer strains [119]. Bicornutin is an arginine-rich linear hexapeptide produced as a complex of several molecules with consensus (RLRRRX) with potent activity against plant pathogens such as *Erwinia amilovora* and the oomycete *Phytophtora nicotianae* [132].

Nematophin (Figure 5) is a small indole antibiotic and antifungal compound first reported in *X. nematophyla* in 1997 whose NRPS biosynthetic origin was finally characterized in 2017, alongside other analogues from different *Xenorhabdus* species found through genome mining. Nematophins are structurally related to xenortides and rhabdopeptides, both very important antiprotozoal and cytotoxic compounds [126,133]. While xenortides were originally discovered in *X. nematophyla* [126] rhabdopeptides are widespread in both *Xenorhabdus* and *Photorhabdus* strains, in the latter mostly represented by the mevalagmapeptides (Figure 5) [120,133,134]. Rhabdopeptides are linear molecules containing two to seven amino acid residues, mainly valine, phenylalanine and leucine (often N-methylated) fused to a terminal amine, which is variable in structure and usually depends on the pool of available amines in each producer strain [134,135]. Interestingly, these molecules are synthesized by NRPS systems consisting of just two to three modules encoded in stand-alone proteins. These modules can then act iteratively adding several units of the same amino acid and interact with each other in a combinatorial way, which allows the generation of libraries of multiple peptide products with different lengths and compositions, all starting from a limited set of simple elements [134]. Further screening in other strains has therefore the potential to reveal even more representatives of this highly bioactive family of molecules [135]. PAX peptides (Peptide-Antibiotic-Xenorhabdus) are lysine rich cyclic lipopeptides exclusive to *Xenorhabdus* with antibiotic and strong antifungal activity that are also produced in sets of several analogues [136,137] whereas gameXpeptides (Figure 6), very widespread both in *Xenorhabdus* and *Photorhabdus* are a family of cyclic pentapeptides with no known function [138,139]. Untargeted metabolomic analyses coupled with mass spectral networking has revealed subfamilies of cyclic peptides containing three to six amino acids structurally related to gameXpeptides. The fact that these molecules seem to be one of the few widely conserved within the *Xenorhabuds*/*Photorhabdus* clade and that those strains that do not produce them produce instead another novel family of hydrophobic depsipeptides termed xefoampeptides suggests these molecules must have a relevant role in the biology of these bacteria [119]. Other relevant NRPS-derived molecules in *Xenorhabdus* are xenorhabdins and xenorxides (Figure 6), both belonging to the category of dithiolopyrrolone antibiotics and with strong antibacterial and cytotoxic activities [140] and recently identified odilorhabdins, broad spectrum antimicrobials with activity against drug-resistant Gram-negative bacteria and with proven efficacy in animal models of infection [141,142].

Interestingly in *Photorhabdus*, mevalagmapeptides, gameXpeptides and several other metabolites such as the giant pentadecapeptide NRP kolossin were only identified through promoter exchange, placing their gene clusters under the control of the arabinose inducible *P*_BAD_/AraC system [120]. The same approach led to the identification of cyclic peptide photoditritide in *P. temperata* Meg1 [143], whereas the antiprotozoal cyclic lipopeptide phototemtide A (Figure 6), originally from the same strain, was identified after heterologous expression of its cluster in *E. coli* [144]. Other group of molecules characterized by a combination of comparative genomics analysis and heterologous expression in combination with protease inhibitor treatment are the bovienimides, lipotripeptides carrying a C-terminal D-citrulline residue (Figure 6) [145].

Another big set of metabolites are the product of PKS-NRPS systems. That is the case for xenocoumacins, fabclavines, pristinamycin, or glidobactin in *Photorhabdus*. Xenocoumacins (Figure 6) are perhaps the main antimicrobial metabolites from *Xenorhabdus*, exhibiting broad spectrum activity against Gram-positive and Gram-negative bacteria, as well as some fungal species [146]. These molecules are characterized by the presence of a PKS-derived isocoumarin group fused to leucine and arginine or leucine and proline residues in xenocoumacin 1 and 2, respectively [146,147]. Further molecules belonging to this group have been identified varying culture conditions, from modifying fermentation pH, to using culture media mimicking the amino acid composition of the insect haemolymph [146,148]. Fabclavins are another family of hybrid molecules that, in addition to the PKS and NRPS-derived elements, present a polyamino moiety derived from polyunsaturated fatty acid-like biosynthesis. These molecules also exhibit a broad target spectrum all the way from bacteria to eukaryotes, and have been postulated as protection compounds against food competitors of *Xenorhabdus* [149]. Activation of the highly conserved xenocoumacin BGCs in diverse *Xenorhabdus* species via integration of chemically inducible promoters has led to the recent identification of over 20 more members of this family [141]. Pristinamycins, originally thought to be produced only by *Streptomyces*, have also been detected in *X. nematophyla* cultures, after finding its BGC through comparative genomics. Interestingly, it seems this species might have acquired the BGC through horizontal gene transfer [150]. Glidobactins and related molecules cepafungins (Figure 6) and luminmycins are potent proteasome inhibitors produced by certain *Photorhabdus* species, initially detected through screening of different culture media and fermentation conditions, as well as during infection of live insects [151,152]. The use of heterologous expression approaches as well as cryptic cluster activation through promoter insertion in combination with advanced metabolomic analyses have allowed the identification of further molecules belonging to this family [153,154].

While NRPS-derived peptides are the major products of the Xenorhabdus/Photorhabdus clade, these bacteria have also been reported to produce ribosomal peptides. Xenocin, produced by *X. nematophyla*, is an RNase delivered through a type III secretion system and produced under iron deprivation conditions [155,156]. Xenorhabdicin, also the product of several *Xenorhabdus* species, is an R-type or phage tail-like bacteriocin that targets other *Xenorhabdus* and *Photorhabdus* species [157,158,159]. Together with xenocin they are proposed to inhibit competition from closely related species in the iron-deficient larval environment [156]. More recently a bioactivity screening of concentrated extracts from several *Photorhabdus* and *Xenorhabdus* strains revealed the production of a novel peptide molecule by *Photorhabdus khanii* HGB1456. This molecule, named darobactin (Figure 6), is a modified heptapeptide containing two macrocycle crosslinks and is the product of a RiPP BGC containing a radical SAM SPASM protein [160]. Darobactin selectively kills Gram-negative bacteria by attacking a novel target, the outer membrane protein BamA. Genomic studies of the BGC revealed this cluster is spread across *Photorhabdus* species, as well as other animal-associated bacteria such as *Yersinia pestis*, among others [160].

Despite the already thorough exploration of the biosynthetic potential of *Xenorhabdus* and *Photorhabdus* it is clear that further screening of novel strains in this group as well as the application of modern genomics and metabolomics techniques are still successful in the discovery of novel antimicrobial metabolites from these bacteria [119,161]. However, these are not the only relevant entomopathogenic bacteria to investigate in the search for new natural products. *Paenibacillus*, a genus of Gram-positive endospore-forming bacteria (phylum Firmicutes), is a promising secondary metabolite source [162,163]. This genus is not a specialized entomopathogen, as *Xenorhabdus* and *Photorhabdus* are, and in fact numerous representatives have been reported as plant growth promoters through nitrogen fixation, production of phytohormones or immunity stimulation [162]. On the other hand *Paenibacillus larvae* is well known as the causative agent of the American Foulbrood disease in honeybees and *P. glabratella* infects snails, whereas other species are opportunistic human pathogens [162]. From a secondary metabolism point of view, *P. polymyxa* is renowned as source of natural products, in particular polymyxins. These are NRPS-derived cyclic lipopeptides that interact with lipid A in the outer membrane of Gram-negative bacteria [164,165]. Polymyxin B (Figure 7) and E (also known as colistin) are used as last resource antibiotics for the treatment of multidrug resistant Gram-negative pathogens. Discovered in the 1940s, they fell in disuse due to their toxicity, but the current antimicrobial resistance crisis and the lack of alternatives have quickly brought them back to relevance [165]. In addition to polymyxins, *P. polymyxa* strains also produce fusaricidins, hexadepsipeptides with a lipid side-chain active against Gram-positive bacteria, pathogenic fungi such as *Fusarium* or even the closely related *P. larvae*, as well as (L)-(−)-3-phenyllactic acid (Figure 6), another antifungal molecule [166,167,168]. A recent comparative genomic analysis of over 40 *P. polymyxa* strains has revealed this species is biosynthetically gifted, with its pangenome containing hundreds of potential BGCs, many of those with no known products [169]. Besides these, diverse *Paenibacillus* species produce multiple antimicrobial peptide molecules like the macrolactone peptide antibiotics paenialvins [170], iturin-like paenilarvins (Figure 7) [171], linear lipopeptides-like tridecaptins [163,172] and saltavalin [173] or cyclic cationic lipopeptides, including the polymyxin-like octapeptins [174,175], pelgipeptins (Figure 7) [176], gavaserin [173] or paenibacterin [177,178,179], among many others [163,180]. *Paenibacillus* also produce RiPPs, such as the lantibiotics paenibacillin [181,182], paenicidins A [183,184] and B, penisin [185] or the class II bacteriocins pediocins [162]. While less abundant, *Paenibacillus* also contain PKS and hybrid PKS-NRPS clusters, such as the ones responsible for the biosynthesis of paenamacrolidin in *Paenibacillus* sp. F6-B70 and paenilamicin in *P. larvae*, respectively. Paenimacrolidin (Figure 7) is a macrolide with a 22-membered ring carrying side chains, with reported activity against methicillin-resistant *S. aureus*. While its BGC has yet to be confirmed, genomic and functional studies strongly suggest this is the product of a trans-AT PKS system. On the other hand, the hybrid biosynthetic origin of paenilamicin (Figure 7) has been well characterized and this compound exhibits antimicrobial, antifungal and cytotoxic activity, leading to the hypothesis that it could contribute to the virulence of *P. larvae* [186,187]. In summary, *Paenibacillus* can be considered a gifted genus, not only for the amount of natural products their species produce, but also because most of these have antimicrobial or antifungal activities [162,163,180].

Another gifted genus often associated with insect pathogenicity is *Serratia*. *Serratia entomophila* is toxic to larvae of several species of *Phyllophaga* (scarabs) among other insects and more recently several strains of *Serratia marcescens* have been reported to have the same effect [188]. *Serratia marcescens* has been better studied for its biosynthetic potential and produces numerous metabolites such as the pigment prodigiosin, siderophore serratiachelin or the macrolide serratamolide (Figure 8) among others. A recent untargeted metabolomics and molecular networking approach has expanded the chemical diversity in this species, by finding 18 novel analogues of these molecules [189]. A related species, *Serratia plymuthica*, is also a promising candidate for screening as the producer of serratamid (Figure 8), a hybrid PK-NRP molecule with activity against phytopathogenic bacteria such as *Ralstonia* and *Xanthomonas*, as well as several different siderophores previously reported in other Gram-negative bacteria [190,191]. The examples reported here represent just the best-studied, well-known entomopathogenic bacteria, but exploration of other entomopathogens and, by extension, pathogenic bacteria in general, who often live in very competitive environments, could lead to the discovery of novel antimicrobial compounds [192].

## 5. Anaerobes

In recent years, anaerobic bacteria, and especially those belonging to *Clostridium* (phylum Firmicutes) and related genera have been revealed as a previously overlooked but very promising source of new specialized metabolites. The analysis of their genomes has identified BGCs for the biosynthesis of different classes of secondary metabolites, and experimental approaches have confirmed that potential with the discovery of several compounds of outstanding interest.

The first and perhaps most fascinating case refers to the discovery of closthioamide (Figure 9) [193] from *Ruminiclostridium cellulolyticum*. Closthioamide is a hexathioamidated antibiotic [194] with a completely unprecedented structure. Despite being a non-ribosomal peptide (NRP), closthioamide is not the product of a NRPS assembly line. Instead, its amidated backbone is made employing an unusual thiotemplated strategy in which the amide bonds are synthesized by ATP-grasp and cysteine protease proteins [195,196]. This illustrates how the production of peptide secondary metabolites can follow patterns beyond the classic NRPS and RiPP paradigms. In addition to this, the biosynthesis of closthioamide also involves a new thioamidation mechanism, based on the action of a thioamide synthetase belonging to the adenine nucleotide α-hydrolase protein superfamily on a thiotemplated substrate [197], making the biosynthesis of closthioamide one of the most novel and exciting pathways described in recent years [198]. Bioinformatics analyses using the unusual closthioamide biosynthetic enzymes as baits have revealed a previously unknown diversity of NRPS-independent NRP pathways waiting to be unveiled [196,197]. Interestingly, *R. cellulolyticum* was originally isolated from decayed grass compost, and the production of closthioamide during fermentations could only be triggered by the addition of soil extracts to the culture media [193]. Special culture conditions seem to be usually required for the production of secondary metabolites by anaerobes, as indicated by years of unsuccessful search for the products of any of the numerous BGCs treasured within their genomes [199,200]. The low energetic efficiency of anaerobic metabolism is a likely reason for that strict control of the secondary metabolism, which normally demands a lot of energy. Thus, anaerobes only seem to produce specialized metabolites under the precise conditions in which they are required. Another interesting example of this is clostrisulfone (Figure 9), a diaryl sulfone from *Clostridium acetobutylicum* whose production is triggered by the addition of supraphysiological concentrations of cysteine to the fermentation medium [201].

Clostrubin A (Figure 9), isolated from the industrially relevant *Clostridium beijerinckii*, has been the first polyketide discovered from an anaerobe microorganism, and exhibits potent antibiotic activity against important nosocomial pathogens [202]. This compound features an unprecedented pentacyclic polyphenol structure originated by a type II PKS system [203], a biochemical feature exceptionally rare in non-actinobacterial microorganisms. Even more interestingly, clostrubin B, produced by the potato pathogen *Clostridium puniceum*, plays an exceptionally important dual ecological role: its antibiotic activity reduces competition from other microorganisms during the tuber colonization and at the same time confers oxygen tolerance to its producer, which cannot grow under oxygenic conditions in the absence of this compound [204].

## 6. Myxobacteria

While many other groups of bacteria have been neglected for a long time as possible sources of new natural products, Myxobacteria (class Deltaproteobacteria, order Myxococcales) are a well-known source of chemical diversity, and their secondary metabolism possibly constitutes one of the most studied after Actinomycetales [205,206]. Myxobacteria inhabit mainly the soil, but lately they have been also isolated from marine habitats [207], and they undergo complex life cycles involving free cell forms, and social swarming aggregation and formation of fruiting bodies when resources are scarce [208]. They harbour impressively large genomes, including the biggest among bacteria (with a record of 14.7 Mb held by a *Sorangium celollosum* strain [209]). These gigantic genomes encode pathways to produce a wide variety of specialized metabolites, with an abundance of BGCs comparable to Actinomycetales [210]. The compounds produced by myxobacteria seem to differ remarkably from the ones produced by other microorganisms, with more than 40% of the myxobacterial compounds being structurally novel and, in many cases, carrying unique chemical features. Interestingly, their chemical space exhibits a very high proportion of hybrid PK-NRP compounds, in contrast with the dominance of pure PKS or NRPS pathways found in Actinobacteria [211]. The trend to combine different classes of biochemical pathways to generate chemical diversity seems to be general among this group of bacteria, as illustrated by leupyrrins (Figure 10), which combine within their structures moieties originated by PKS, NRPS and terpene biosynthetic machineries [212,213]. The potentially antithrombotic myxadazoles (Figure 10) are unprecedented hybrid compounds, combining a benzimidazole moiety (derived from vitamin B12 metabolism) with a linear fatty acid (of PKS-NRPS origin) endowed with an isoxazole ring [214]. Remarkably, the NRPS and PKS assembly lines found in myxobacteria often diverge from the canonical models found in other bacteria in terms of module architectures, biochemical behavior, building block selection and chain termination [215]. As a result, the final products of the pathways include in many cases highly unusual moieties, as illustrated by the alternative heterocycles featured by the myxofacyclines, including isoxazole, 4-pyrimidinolide and 1,2-dihydropyrol-3-one, all of them generated by the same PKS-NRPS system [216]. In addition to this, the molecular targets for myxobacterial compounds are in many cases rarely targeted by compounds produced by other microorganisms [211]. This is, for example, the case for soraphen A (Figure 10) [217,218], a type I PKS product that, at nanomolar concentrations, inhibits the biotin carboxylase domain of the eukaryotic acetyl-CoA carboxylase, a completely novel mode of action with potential implications for the development of anticancer agents. From a pharmacological perspective, the most important myxobacterial-derived compounds discovered so far are epothilones (Figure 10), PKS-NRPS hybrid products that act as microtubule-stabilizing agents [219,220]. This mode of action, akin to taxol, provides these compounds with a great potential as anticancer agents. In 2007 the FDA approved a semisynthetic epothilone derivative (ixabepilone, marketed by BMS) for the treatment of metastatic breast cancer.

Myxobacteria have a predatory lifestyle, and it is possible that a good part of their exotic chemical repertoire could be indeed related to predation [221,222,223]. This idea is reinforced by the fact that many of their secondary metabolites are overproduced during exponential growth phase [224], in contrast with the regulation of the secondary metabolism in Actinobacteria, which normally is activated during the stationary phase for fitness improvement under nutrient-starvation conditions [225]. However, other myxobacterial BGCs are cryptic, as in the case of the plasmid-encoded sandarazols (Figure 10) from *Sandaracinum* sp. Msr10575, a recently described and structurally unprecedented group of toxins or defensive compounds where biosynthesis could only be activated by promoter exchange in the native producer [226]. Sandarazols contain within their structures a series of highly reactive functional groups, including an α-chlorinated ketone, an epoxyketone and (2R)-2-amino-3-(N,N-dimethylamino)-propionic acid building block. Heterologous expression is also a useful alternative for production of myxobacterial compounds, as probed by using *Pseudomonas putida* as the expression host [227]. Extensive reviews about the natural products from Myxobacteria have been published [228,229], but the great diversity of this group of bacteria and the abundance of unusual BGCs within their large genomes guarantees fascinating future discoveries [230,231].

## 7. Cyanobacteria

Cyanobacteria, the phylum of Gram-negative, slow-growing photosynthetic bacteria, are another prolific source of natural products. This ancient lineage of bacteria is widely distributed in nature and can be found both in marine and freshwater bodies, as well as in the soil and some extreme environments or even in symbiotic associations, such as lichens [232,233] or with multiple marine invertebrates, as previously discussed. From the secondary metabolism point of view, cyanobacteria are arguably the second most studied group after Actinobacteria, mainly due to their ecological impact and human health and economic implications. Algal blooms, with the ensuing production of toxins, are an ever-increasing occurrence with lethal consequences for aquatic populations and also for humans, who can become exposed to cyanotoxins through dermatological contact or ingestion of contaminated seafood and water [234,235,236,237]. As a result, cyanobacterial toxins have been extensively studied for over 50 years, with saxitoxin, one of their best-known representatives (also produced by dinoflagellates), structurally characterized back in the 1970s [238,239]. To date, more than 2000 cyanobacterial natural products have been characterized, most of them with some kind of biological activity, ranging from the aforementioned toxins to antibacterial, antifungal, antiprotozoal, antiviral, antialgal, anti-inflammatory, antioxidant or protease inhibition, among others. For a comprehensive overview and catalogue of cyanobacterial natural products as well as their bioactivities and biosynthetic origins, please refer to recent reviews and resources focused on this group of microorganisms [232,240,241].

Despite their proven biosynthetic potential, a truly systematic study of cyanobacterial secondary metabolism has traditionally been hindered by several aspects of their biology. On one hand, cyanobacterial natural communities tend to be very complex, often establishing symbiotic relationships among the members of the population, which makes obtaining axenic cultures of a specific strain highly challenging. As a result, natural product characterization has been frequently carried out in assemblages, bacterial consortia where the actual producing organism is not known [77,242,243]. This has been overcome to some extent by the advent of next-generation genome sequencing techniques, in particular metagenomics and single-cell sequencing, which allows the unpicking of the bacterial composition of complex bacterial communities, including unculturable bacteria [244,245,246]. In fact, it has been shown that some compounds originally thought to be produced by higher organisms are instead products of cyanobacterial origin, acquired through feeding and symbiotic interactions (as in the case of patellamides in certain tunicates or dolastatins in molluscs, see above). Another challenge in the study of cyanobacterial natural products is the genetic manipulation of their producers. Cyanobacteria are polyploid and each cell carries a variable number of copies of their genome, complicating the segregation of desired genotypes after manipulation [247,248]. In addition, until quite recently there were few model cyanobacteria suitable as hosts for heterologous expression or genetic toolkits akin to those available in Actinomycetes. Often genetic work on cyanobacterial natural products was carried out through cloning of genomic or metagenomic fragments of interest and their expression in other organisms such as *E. coli* [249,250,251]. More recently, filamentous cyanobacteria like *Anabaena* PCC 7120 and unicellular cyanobacteria with fast doubling times and good biomass production, such as *Synechococcus elongatus* PCC 7942 or *Synechococcus* sp. PCC 11901, are being explored as heterologous expression hosts [233,252,253,254,255]. In parallel, multiple genetic manipulation tools for cyanobacteria are being developed, including modular vector systems such as *Synebrick* for *S. elongatus* PCC 7942 [256] or the GoldenGate moclo-based CyanoGate system [257], CRISPR-Cas-based tools [258] as well as multiple synthetic biology elements and other techniques to optimize gene cluster expression [233,259].

As with many other microorganisms, recent sequencing efforts have revealed that the biosynthetic potential of cyanobacteria has been underestimated. Over 70% of the compounds discovered to date come from bacteria from the orders Oscillatoriales and Nostocales, followed by Chroococcales and Synechococcales. The rest of the orders in this phylum have very few secondary metabolites associated, but a genomics study of a collection of taxonomically diverse cyanobacteria has revealed that 70% of the strains sequenced do contain biosynthetic gene clusters [232,260]. The type of cluster and distribution changes between different groups of cyanobacteria, suggesting that surveys of taxonomically diverse bacteria could reveal more biosynthetic diversity [260]. On the other hand, genomic studies of well-known secondary metabolite producers, such as the genus *Moorea* (previously included within the *Lyngbya* polyphyletic group) indicate that these bacteria contain between 30 and 45 BGCs and can devote up to 20% of their genomes to secondary metabolism [261], highlighting their still underexplored potential.

Cyanobacterial metabolites are chemically diverse and include polyketides, terpenes, alkaloids, as well as lipids and polysaccharides, but these organisms are particularly talented producers of peptide or peptide-derived molecules, in a similar way to what was observed for *Photorhabdus* and *Xenorhabdus* [232]. It is estimated that over 60% of all cyanobacterial natural products are peptides in nature, derived from NRPS, hybrid NRPS-PKS systems and RiPP BGCs [232,233,240]. This is partly due to the higher abundance of these clusters in cyanobacteria, but also to the fact each cluster is often capable of producing several different congeners of the same kind of molecule, due to the substrate flexibility of NRPS loading domains and precursor peptide redundancy and repetition in the case of RiPPs [240,262]. As a result, the secondary metabolite repertoire of cyanobacteria expands greatly with any newly characterized BGC. As previously noted, the best-known metabolites from cyanobacteria are usually cyanotoxins, which have diverse biosynthetic origins. Microcystins (Figure 11) and the closely related nodularins, both highly hepatoxic fresh water cyanotoxins are cyclic heptapeptides and pentapeptides, respectively, produced by a hybrid PKS-NRPS system, with the PKS component responsible for the synthesis of the rare β-amino-pentaketide ADDA [263,264]. The NRPS component exhibits substrate flexibility, leading to the generation of multiple analogues per cluster. This, in addition to the screening for these kind of clusters in different cyanobacteria, has revealed well over 100 different microcystin-like toxins. As recently as 2020 a study combining LC-MS and ELISA immunoassay screenings uncovered more than 70 microcystins being produced by just two *Microcystis* strains, 22 of those completely novel [265]. Cylindrospermopsin (Figure 11), another hepatotoxin, is instead an alkaloid molecule, also produced by a PKS-NRPS system [265,266]. Other examples of cyanobacterial molecules produced by hybrid systems are barbamides [267] (Figure 11), nostopeptolides [268], microginins, which exhibit multiple clinically interesting bioactivities [269,270], laxaphicins, with synergistic cytotoxic and antifungal activity [271,272], or the recently characterized cryptomaldamides (Figure 11), discovered through matrix-assisted laser desorption/ionization (MALDI) analyses of individual filaments of *Moorea producens* fed with heavy nitrogen and whose BGC was later cloned and heterologously expressed in *Anabaena* PCC7120 for further characterization [254,273]. On the other hand, molecules like aeruginosins, anabaenopeptin (Figure 11), nostocyclopeptides [274,275], the antifungal cyclic lipopeptide hassalidins [276,277] among others, are the product of NRPS-only clusters. Anatoxin-a (Figure 11) and saxitoxin are potent alkaloid neurotoxins derived from PKS BGCs instead [278,279]. New variants of these molecules keep being discovered with updated LC-MS-based screening and targeted PCR screening of biosynthetic genes [279,280,281]. Another very relevant example of PKS-derived cyanobacterial molecules is the swinholide-like family of macrolides, with more than 60 representatives including samholides, scytophycins (Figure 12), ankaraholides, among many others. These molecules, with promising cytotoxic activity, are the product of trans-AT PKS systems which were initially found in the genomes of heterotrophic symbionts of marine invertebrates [282,283,284], as previously discussed.

RiPPs are also a rapidly expanding class of natural products prevalent in cyanobacteria, with cyanobactins and microviridins being the best-known representatives. Cyanobactins, which include molecules such as the previously described patellamides (Figure 12), aeruginosamides, anacyclamides or trunkamides, are a very diverse group of RiPPs but they all present a heterocycle in their C-terminus, either a proline or an azol(in)e installed by YcaO heterocyclases, and most of them are cyclized, due to ligation between their N- and C-termini [262,285]. In addition to that, they can carry many other post-transcriptional modifications, including further heterocyclizations, methylations or prenylations, to name a few. To date, more than 100 different cyanobactins have been characterized [285,286,287]. This level of diversity is due to the widespread distribution of these BGCs among cyanobacteria, but also to the presence of multiple core sequences, arranged in cassettes within the precursor peptide in each cluster [285]. Microviridins (Figure 11), now integrated in the graspeptide family [39], are tricyclic-acetylated depsipeptides, potent inhibitors of serine proteases. The intramolecular linkage is catalyzed by ATP-grasp ligases, signature tailoring enzymes in these molecules BGCs [288,289]. Another highly diverse family of RiPPs is that of lanthipeptides such as prochlorosins, which are remarkable for the extreme substrate flexibility of their lanthionine synthetase, capable of modifying dozens of structurally distinct precursor peptides [290,291]. Genome mining and untargeted metabolomics efforts have revealed many representatives of these kind of molecules, proving the power of multiomics approaches to uncover further RiPP diversity [289,292]. This is also how other RiPP families, like proteusins and spliceotides, were identified in cyanobacteria. Proteusins are a family of RiPPs whose precursor peptides present a nitrile hydratase domain in their leader region. The first representative of this family was polytheonamide, from an uncultivated sponge symbiont (see above), but a genomic survey of cyanobacterial genomes revealed the presence of over 50 proteusin-like BGCs in cyanobacteria, many of them with multiple precursor peptides. Expression of one of these clusters in *E. coli* resulted into the production of landornamides (Figure 12), the first characterized cyanobaterial proteusin [293]. Similarly, the spliceotide family was identified in cyanobacteria after genomic studies revealed the presence of clusters containing nif11 domain-containing precursor peptides, associated with orphan radical SAM enzymes. No naturally occurring spliceotides have been isolated to date and there is no knowledge about their potential biological role, but coexpression in *E. coli* of these nif11 precursor peptides with their cognate radical SAM enzymes revealed their novel enzymatic activity, splicing the core peptide at one of the amide bonds and removing a tyramine equivalent to generate a β-amino acid in that position [294]. These results, alongside other genome mining studies revealing the presence of multiple RiPP cluster families, several of them still uncharacterized [295], indicate that there is plenty of cyanobacterial “chemical dark matter” still available to explore.

## 8. Lichens

As briefly mentioned before, cyanobacteria often establish symbiotic relationships with the organisms they share a habitat with. Arguably the best-known case is that of lichens, structures traditionally described as the result from the symbiosis between heterotrophic filamentous fungi (mycobionts) and photosynthetic algae or cyanobacteria (photobionts). This symbiosis is ancient and involves extensive physiological interdependencies, and while it is sometimes possible to obtain axenic cultures of the photobiont partner, the mycobionts are often refractory to in vitro propagation or isolation [284,296]. Metagenomic and metatranscriptomic approaches have therefore been instrumental in the characterization of the different partners in these associations and their biosynthetic potential [296,297,298]. They have also revealed that lichens are in fact better described as micro-communities with multiple additional components, such as basidiomycete yeasts and other bacteria, mainly Proteobacteria and Actinobacteria [299,300,301]. Lichens are widespread in nature and can be found in multiple habitats, including extreme environments [302,303], and it is thought the mycobiont offers structural support and protection against abiotic stress, whereas the photobiont offers nutrient acquisition through photosynthesis and nitrogen fixation [299].

With regards to lichens’ secondary metabolites, both the mycobiont and photobiont partners as well as their associated bacteria have the ability to produce secondary metabolites, multiplying the biosynthetic potential of these organisms [299,304]. Several compounds associated to free-living fungi and especially cyanobacteria can be isolated from lichens, such as the antifungal glycolipopeptides hassalidins [277], nostocyclopeptides [305] or multiple cyanotoxins from the microcystin family [306]. On the other hand, there are a set of metabolites highly specific and prevalent in lichens, such as the families of phenolic compounds depsides (such as the molecule atranorin), depsidones, dibenzofurans and pulvinic acid derivatives [299,307]. Several representatives of these families exhibit multiple biological properties, including antioxidant, antibiotic, antiproliferative or even anticoagulant activity [307,308,309] and have only been reported in lichens or in some plants, as the product of endophytic fungi [310]. The best-known dibenzofuran is usnic acid, a bitter yellow pigment present in multiple lichens and thought to primarily offer photoprotection and defence against predators, but with many other reported biological activities [311]. Finally, pulvinic acid derivatives are also pigments present in lichens but also in some free-living fungi, with proposed antioxidant activity [312]. The presence of related compounds in non-lichenizing fungi suggested these compounds were the product of the lichen mycobiont, a hypothesis that was confirmed through genome sequencing, metagenomics and comparative genomics, along with heterologous expression of candidate producer BGCs [313,314,315,316]. In fact, a wider genomic and bioinformatic analysis of mycobionts from lichens revealed the presence of multiple type I PKS genes, underpinning the potential of these symbioses to specifically produce these kind of molecules [317] New compounds belonging to these families keep being characterized [313,318], but given that fungi and photosynthetic algae are outside of the scope of this review we will not focus further on their products, which have already been described in detail in other works [299]. Coming back to cyanobacteria, it is interesting to note that cyanobacterial photobionts produce some metabolites mainly detected in other symbiotic bacteria, such as the macrolide swinholides or the polyketide nosperin (Figure 13), both from *Nostoc* species [284,296]. The case of nosperin is particularly interesting, as it is the product of a trans AT-PKS system and belongs to the family of pederin compounds, which are almost exclusively produced by endosymbionts of eukaryotic organisms [296]. Even in the case of microcystins, it seems that the molecules produced by cyanobionts are comparatively rarer and present unusual structural modifications with respect to the ones produced by free-living cyanobacteria [299,319]. These observations, alongside other untargeted metabolomics’ studies on lichens strongly suggest that some secondary metabolites can be specific to symbiotic relationships and that exploring these niches can expand the chemical diversity of this kind of metabolites [296,320]. This is true not only of lichens or their main components, as further work on lichen-like symbioses [321] or other lichen associated microorganisms, mainly Actinobacteria [303,304,322], has revealed rich, secondary metabolite-driven interactions [323] and promising leads for clinical use such as the enedine antibiotic uncialamycin (Figure 13).

## 9. Pseudomonas

These rod-shaped, generally motile Gram-negative microorganisms belonging to the Gammaproteobacteria class are perhaps the most cosmopolitan genus of bacteria. With hundreds of known species, *Pseudomonas* strains can be found worldwide and in multitude of different environments, from soil-dwelling, rizhosphere-associated species such as the members from the *Pseudomomas fluorescens* group, to insect-associated ones like *Pseudomonas entomophila* or the opportunistic pathogens *Pseudomonas maltophilia* and *Pseudomas aeruginosa* [324,325,326]. Their relevance to humans is particularly high, not only in the clinic, with *P. aureginosa* belonging to the critical group of ESKAPE pathogens [8], but also in agriculture, with multiple crop diseases caused by different pathovars of *Pseudomnas syringae* [327,328,329] and, conversely, several *Pseudomonas* strains identified as plant growth-promoting bacteria [330,331,332,333]. The explosion in data availability in this next generation sequencing era has revealed that the huge adaptability of these bacteria is directly related to their extreme genomic versatility, with the core of conserved genes within the genus being relatively small compared to the accessory genes that conform its pangenome. Moreover, these accessory genes are specific for different lifestyles and environments, making the pangenome from the *P. fluorescens* group, for example, highly distinct from that of *P. aeruginosa*. Comprehensive sequencing efforts are continuously shedding light on this expanded pangenome, and are helping with the clarification of the rather complicated taxonomy of *Pseudomonas* species [325,326,334]. From the secondary metabolism point of view, *Pseudomonas* are definitely an interesting source of natural products, particularly in the case of those strains with proven beneficial phenotypes [333], as many of the previously mentioned accessory genes are devoted to the biosynthesis of secondary metabolites [326]. Certain types of metabolites, such as siderophores and cyclic lipopeptides seem to be staples of this genus and are widespread in it [326], but there are also other specialized metabolites, many of them common to other often distantly related bacteria, such as *Streptomyces*. This shared metabolic potential can arise through horizontal gene transfer from other environmental bacteria, as seems to be the case for antibiotic byciclomycin [335] or the coronafacoyl phytotoxins [336] but also occurs through convergent evolution of separate pathways as it is the case for antibiotic fosmomycin [337,338] or tropolone siderophores [339,340,341].

Most *Pseudomonas* secondary metabolites are NRPS-derived peptides. This is the case of the pyoverdine family of siderophores, green fluorescent molecules with high affinity for ferric iron that give their characteristic aspect and name to *P. fluorescens*, but are widespread in the genus, including the pathogen *P. aeruginosa*, where they contribute to its virulence [342,343,344]. Pyoverdines (Figure 14) contain a quinoline derivative, responsible for the siderophore pigmentation, attached to a peptide backbone and N-acylated with varying dicarboxylic acids or their amides [326,345]. Each strain is usually capable of producing a set of pyoverdines, with different acyl chains but a constant amino acid backbone. In contrast, this backbone varies from species to species, making pyoverdines so specific that their potential as taxonomic markers has been considered [344,345]. A peculiarity of pyoverdines, in contrast with most natural products described so far, is that their BGCs are often fragmented and dispersed across *Pseudomonas* genomes, sometimes in up to five fragments, an unusual arrangement variability that potentially contributes to their high structural diversity and specificity [326]. Another characteristic group of secondary metabolites from pseudomonads are the lipopeptides. These molecules consist of a peptide backbone of varying lengths acylated in its N-terminus with a linear fatty acid, which can also vary in length [326,346]. While a few of these lipopetides are linear, such as syringafactins [347,348], corrugatins [346], sclerosin [349] or tolaasin C [350], the overwhelming majority of these compounds are cyclic and present a macrolactone ring generated by the condensation of their C-termini with an hydroxyl group from a threonine or a serine from the peptide backbone [346]. *Pseudomonas* cyclic lipopeptides can be subdivided in multiple different groups according to the length of their peptide chain and its composition, and some of these groups include multiple congeners that vary slightly from strain to strain, or even within strains, when their NRPS BGCs have substrate flexibility. Some of the most prevalent cyclic lipopeptides groups include viscosins (Figure 14), syringomycins, amphisins, putisolvins, orfamides or syringopeptins [326,346]. Lipopeptides are amphipathic molecules which can act as biosurfactants, assisting *Pseudomonas* motility and colonization of their environmental niche. In addition, many of these molecules are also antibacterial, antifungal and antiprotozoal and have been shown to contribute to the biocontrol activity of plant growth promoting strains, as well as participate in cooperative predator defence, among many other activities [333,351,352,353]. As more strains are isolated, sequenced and their metabolomes assessed, more of these molecules are discovered [354,355,356]. A recent example of this are bananamides, a novel family of cyclic lipopeptides uncovered by untargeted metabolomics and molecular networking in a collection of 260 pseudomonad strains [354]. Other NRPS-derived molecules from *Pseudomonas* include the siderophores pyochelin (Figure 14) and pseudomonine, as well as molecules like the phytotoxin tabtoxin (Figure 14) [357], the antitumor safracin [358,359], the β-lactone antibiotic obafluorin [360,361,362] or the recently characterized azabicyclene (Figure 14), discovered in *P. aeruginosa* after upregulation of its BGC through feeding with the quorum-sensing signalling molecules acylhomoserine lactones [363].

Although less frequent, *Pseudomonas* also have PKS-derived secondary metabolites, as is the case of the antibiotic mupirocin (Figure 15) [364,365,366] or 2,4-Diacetylphloroglucinol (DAPG), a small molecule which can frequently be found in plant-associated *Pseudomonas* [333,367,368,369], as well as hybrid PKS-NRPS molecules such as the toxins pyoluteorin, syringolin or coronatine (Figure 14) [326,370,371,372]. Interestingly, *Pseudomonas* is also the source of many secondary metabolites with non PKS or NRPS biosynthetic origins, such as the highly diverse and versatile phenazines, which can act as antimicrobials, siderophores and redox acceptors among other functions [373,374,375], the phytohormone Indole-3-acetic acid, hydrogen cyanide [333,376], the recently discovered anti-algal polyyne toxin protegencin (Figure 15) [377,378] or the novel pyonitrins (Figure 15), which are the product of spontaneous condensation of two previously known natural products, pyochelin and pyrrolnitrin, just to name a few [379]. It is in this area, the less easily identified BGCs, where *Pseudomonas* exhibit big potential for novel chemistry discovery. In fact, the use of untargeted methods such as the generation of transposon mutant libraries of bioactive *Pseudomonas* strains has led to the discovery of secondary metabolites such as the butanolide molecules styrolides A (Figure 15) and B [380], or the antimicrobial 7-hydroxytropolone (Figure 15) in two different biocontrol *Pseudomonas* strains [340,341], presenting this as a promising strategy to uncover useful secondary metabolites.

Finally, and perhaps surprisingly, not many RiPPs have been characterized in *Pseudomonas* strains, with the notable exceptions of the redox cofactor pyrroloquinoline quinone (PQQ) (Figure 15) [39] and the recently discovered and highly unusual 3-thiaglutamate (Figure 15) [381]. However, in silico analyses of *Pseudomonas* genomes reveal they indeed contain multiple ribosomal peptide BGCs, often corresponding to bacteriocins, suggesting there is another pocket of secondary metabolite diversity still to be explored in these well-studied bacteria [333,382].

## 10. Burkholderia

*Burkholderia* is a complex (and taxonomically controversial) [383] genus of β-proteobacteria comprising species exhibiting a great ecological diversity, from human pathogens to free-living bacteria, plant pathogens, plant growth promoters and obligate endosymbionts [384]. The clinical relevance of *Burkholderia* has triggered extensive studies on this genus, leading to the accumulation of genome sequences in databases. Interestingly, those genomes (quite large, with an average size of 7.5 Mb) are normally split into two different chromosomes, one of them mainly carrying information for essential cellular functions and a second one which seems to be more enriched in BGCs related to the lifestyle of each strain [385]. *Burkholderia* genomes encode an abundant number of BGCs, varying from 0.8 to 2.2 per Mb [384]. One of the most remarkable aspects of these clusters is the abundance of non-canonical features found in PKS, NRPS and PKS-NRPS hybrid systems. Just to give a few examples of this, the biosynthesis of icosalide A (Figure 16) by *Burkholderia gladioli* requires a very unusual NRPS module arrangement, with two starting condensation domains in two different NRPS modules, in order to generate this diacylated depsipeptide [386]. Icosalide A had originally been isolated from a fungal source, but additional investigations proved that the source of the compound were in fact *B. gladioli* fungal endosymbionts. A similar symbiotic relationship had been observed before in the case of the antimitotic trans-AT PKS-NRPS product rhizoxin (Figure 16) [387], a compound employed by fungal phytopathogenic *Rhizopus* strains in the beginning of the plant infection that generates rice seedling blight. The discovery of the culturable *Burkholderia rhizoxina* endosymbiont as the rhizoxin producer was completely unexpected, given that fungi are by themselves rich sources of secondary metabolites [388,389]. The trans-AT PKS part of the rhizoxin hybrid assembly line includes an unprecedented branching module responsible for the introduction of a δ-lactone side chain [390]. To add another layer of intertwining to this symbiosis, the last epoxidation step in the biosynthesis of rhizoxin is catalyzed by a fungal enzyme [391]. A very unusual combination of cis-AT and trans-AT PKS systems is responsible for the biosynthesis of enacyloxins (Figure 16) in *Burkholderia cepacia* [392], and a novel PKS chain-release mechanism based in an AfsA-like domain and leading to the formation of a butanolide moiety has been identified in the biosynthesis of gladiostatin (Figure 16), a cytotoxic glutarimide compound produced by *B. gladioli* [393]. Other interesting compounds produced by *Burkholderia* are the acetylenic antibiotics cepacins (Figure 16) [394], recently identified as bioactive compounds against plant oomycetal infections [395]. The diversity of specialized metabolites from *Burkholderia* (and other Gram-negative bacteria) has been thoroughly reviewed [384,396,397].

## 11. Planctomycetes

Slowly, more and more groups of Bacteria are being unveiled as potential sources of natural products. In many cases, the establishment of axenic culture conditions for them is the fundamental challenge to solve before accessing their chemical diversity. Planctomycetes constitute an excellent example of this. Planctomycetes are a very distinctive phylum of Bacteria, peculiar in such manner that during a long time it was speculated they could represent an intermediate state between prokaryotic and eukaryotic organisms [398]. However, new discoveries have pointed out that, despite their exceptionality, Planctomycetes are a deep-branching group of Gram-negative bacteria [399,400,401,402]. These microorganisms have a tremendous ecological and biotechnological importance, as a subgroup of them includes the only known organisms able to perform the anaerobic ammonium oxidation (anammox) [403], a metabolic process through which ammonia can be oxidized to dinitrogen in the absence of oxygen. Anammox plays a major role in the global nitrogen cycle and has important applications in nitrogen-rich wastewater management. Planctomycetes can inhabit different ecological niches, but they seem particularly predominant in aquatic habitats. Despite their slow growth, they represent a very substantial part of the microbial biomass in certain ecological niches, rising questions about the possible use of secondary metabolites for competition against other microorganisms. The large size of some of their genomes (up to 11 Mb, median of 7.2 Mb) [404] is also suggestive of a very promising metabolic potential, which is largely unknown: 35–65% of the proteins encoded within them have unpredicted functions [14], a larger proportion than in any other known bacterial lineage. Although initial studies based on metagenomic approaches started pointing out the presence of BGCs within Planctomycetes genomes [405], only a series of recent efforts have managed to isolate and axenically cultivate a number of Planctomycetes strains large enough to provide a general view of the impressive genomic diversity of these bacteria [404]. The conventional bioinformatic analysis of these genomes has revealed a moderate average number of BGCs per genome (from none to 13), but most of them completely unrelated to BGCs from other phylogenetic groups [404,406]. It must also be considered that, given the high proportion of hypothetical proteins present in Planctomycetes genomes, they could hide completely unknown pathways for the biosynthesis of specialized metabolites.

Only relatively simple specialized metabolites from Planctomycetes have been identified so far. The first one, 3,5-dibromo-p-anisic acid (Figure 17) was isolated from Planctomycetes’ enriched cultures and seems to act as a plant toxin, being probably related to the colonization of algal surfaces by the producing strain [407]. The second group of compounds are the stieleriacines (Figure 17) [408], N-acylated tyrosines from *Stieleria neptunia* which have been demonstrated to play an important role in the alteration of the species composition of the marine biofilms inhabited by this bacterium [409]. This finding supports the initial assumptions regarding the importance that secondary metabolism should have for the competition of the slow-growing Planctomycetes within complex microbial communities.

## 12. The Mammalian Gut Microbiome

The mammalian intestine constitutes an impressive anaerobic bioreactor inhabited by an unconceivable number of microorganisms belonging to three domains of life (Bacteria, Archaea and Eukarya), with a number of cells that exceed 10 times the total number of host cells [410]. The gut microbiome plays a wide variety of roles and influences the life of the host in many different aspects, acting in many senses like an additional organ: it stores and distributes energy, performs a great number of metabolic transformations and maintains constant communication with the surrounding tissues. The gut bacterial community reaches cell densities higher than in any other known ecological niche [411]. Despite displaying a relatively low diversity at the phylum level (from the more than 70 accepted bacterial phyla, only eight are represented in the mammalian gut, with Bacteroides, Firmicutes, and to a less extent Proteobacteria being the predominant ones) [412], the gut microbiome is tremendously diverse at the species (hundreds) and strain (thousands) levels. As bacteria–bacteria and bacteria–host interactions are frequently mediated by small molecules, it is easy to imagine that a community of such complexity as the gut microbiome would be a rich source of bioactive-specialized metabolites. This was initially confirmed in the context of a wide exploration of the biosynthetic potential of 2430 bacterial reference genomes from different niches in the human microbiome, which revealed the oral cavity and the gut as the richest sources of secondary metabolites [413], with BGCs for saccharides being the most prevalent, especially in the Bacteroides genomes. RiPPs seem to be widely distributed and slightly enriched with respect to NRPS and PKS BGCs (which are not particularly abundant), but given the particular challenges presented by the bioinformatic predictions of RiPP BGCs [24], it is easy to assume that they could be even more diverse and abundant. Indeed, most of the currently known secondary metabolites from the human gut are RiPPs, such as the lantipeptides ruminococcin A [414], Nisin O [415] and Nisin P [416] or the lassopeptide microcin J25 [417]. However, the most interesting example of gut-derived natural products perhaps corresponds to colibactins (Figure 18), hybrid PK-NRP compounds produced by certain strains of *Escherichia coli* and other proteobacteria. Colibactin features an unusual electrophilic cyclopropane moiety synthesized by an unprecedented S-adenosylmethione-dependent NRPS module [418] and induces double-strand breaks and DNA interstrand crosslinks in the genome of eukaryotic cells, acting as a genotoxin that could be involved in the development of colorectal cancer [419,420,421]. Tilivalline (Figure 18), from *Klebsiella oxytoca*, constitutes another example of how the microbiome activity can be detrimental for the host, as this compound causes the pathogenesis of colitis. Strikingly, tilivalline is the product of the non-enzymatic reaction of tilimycin (the product of a NRPS assembly line) (Figure 18) with biogenetically generated indol. While tilimycin is a genotoxin compound inducing DNA double-strand breaks, tilivalline binds to microtubules arresting mitosis [422,423].

The difficulty of establishing culture conditions for many gut bacteria is remarkable and normally requires high levels of optimization, leading to the development of several in vitro gut models and technologies to facilitate the study of the metabolism of these microorganisms [424]. However, and although extremely valuable when successful, culture-dependent methods can be extremely time consuming an metagenomic and metabolomic methods are frequently employed to access the metabolic diversity of the gut microbiome [425].

## 13. Culturing the Unculturable Treasures from the Soil

We have mentioned several times the difficulties related to the cultivation of many bacteria in the laboratory. Although the reasons for this might be obvious in the case of symbiotic bacteria, the reasons why 99% of the environmental bacteria cannot be grown on synthetic media are not well understood, but they are probably related to growth factor-mediated interdependencies between members of the native microbial communities [426]. Such disparity between the real microbial diversity of natural environments and the culturable microbial diversity constitutes one of the most deeply rooted challenges of microbiology and, although alleviated by the metagenomic technologies, still hampers the study of the diverse specialized metabolism of uncultivated microorganisms. One of the most radical innovations to culture environmental bacteria in isolation has been the development of the iChip technology [15,427], which allows the high-throughput cultivation of 50% of the soil bacteria by the use of diffusion chambers separated from the natural environment by semipermeable membranes that allow the exchange of environmental growth factors. As a striking example of the potential of this technology, this in situ cultivation method allowed the discovery of teixobactin (Figure 19) from a previously unculturable β-proteobacteria, *Eleptheria terrae*, carrying a genome of 6.6 Mb. Teixobactin is a structurally unusual NRPS-derived depsipeptide that represents a novel antibiotic class and targets cell wall biosynthesis by binding to various cell wall precursors, displaying a powerful antibiotic activity against several Gram-positive pathogens [428]. Crucially, since it has a non-protein target, resistance against teixobactin does not seem to develop in vitro More recently, and also employing the iChip technology a *Lysobacter* strain has been identified as the producer of hypeptin (Figure 19) [429], an antibiotic sharing structural features and mode of action with teixobactin, but that was previously described as produced by a *Pseudomonas* strain [430].

## 14. Extremophilic Bacteria

Microorganisms living in extreme environments have since long ago been proposed as potential sources of novel exotic chemistry. It is thought that the stress induced by diverse abiotic factors pushes them to evolve and develop molecules to protect themselves against it. Additionally, the existence of understudied, highly specialized organisms opens the promise of genetic and chemical novelty [302,431]. The main abiotic factors that define extreme environments are temperature, salinity, pH, pressure (especially in the case of deep sea microorganisms) and radiation, solar or otherwise [431,432]. Organisms living in these sort of environments can be classified as extremophilic, if the extreme conditions present are essential for them to live and thrive, or extremotolerant, if, as the name indicates, the organisms can tolerate them but they are not required for their survival [433].

Over the last 20 years, hundreds of secondary metabolites have been isolated from organisms dwelling in hypersaline, arid deserts, permafrost and artic settings, deep sea trenches, hot springs, mine waste pits and many other extreme environments. For an exhaustive report of all the natural products isolated from extremophilic and extremotolerant microorganisms please refer to historic and recent reviews on the topic [302,432,433,434,435,436]. An inspection of the literature reveals that most of the molecules described to date are produced by fungi and Actinobacteria. Cyanobacteria are also a relevant source of molecules in extreme environments, especially photoprotectors such as scytonemin [437,438]. While archaea are prevalent in high-temperature environments, their biosynthetic potential has been reported to be limited to a few NRPS and RiPP BGCs [439]. Besides these, a multitude of unculturable organisms have been identified, many belonging to comparatively rare taxa. While this might have been a limitation for the identification of BGCs in the past, the application of metagenomic techniques should overcome this problem [439,440].

Some examples of molecules produced by non-actinobacterial thermophilic bacteria include ammonificins A to D (Figure 20), hydroxyethylamine chroman derivatives produced by Gram-negative *Thermovibrio ammonificans*, isolated from a hydrothermal vent chimney in the East Pacific [441,442]. Gram-positive *Laceyella sacchari* IT-2L from the family Thermoactinomycetaceae is the producer of bacillamides (Figure 20), N-acetyltryptamine and two N-acylanthranilic acids, of which N-propionylanthranilic acid (Figure 20) was identified for the first time as a natural product [443]. *Thermoactinomyces vulgaris* is the producer of thermoactinoamides A to F, cyclic hexapeptides with activity against *S. aureus* whose BGC has been recently identified [444,445]. *Thermosporothrix hazakaensis* SK20-1, isolated from a field-scale composter is the producer of sattabacin (Figure 20) and hazakacin as well as indole thiazole containing molecules [446,447]. Related compounds 2′-oxosattabacin and ktedonoketone (Figure 20) have been identified as the products of *Thermosporothrix hazakensis* NBRC 105916 [448]. Interestingly this genus belongs to the emerging bacterial class Ktedonobacteria, in the phylum Chloroflexi. This class, originally misclassified as part of Actinobacteria, contains four genera and just six named species, but it is estimated there are many more mesophile representatives to be discovered [449]. A genomic assessment of the few strains from this group sequenced to date revealed the promising, if uneven, biosynthetic potential of these bacteria [450]. In fact, genome mining in one of these strains (*Thermogemmatispora* strain T81, from geothermally-heated soils in New Zealand) lead to the recent discovery of lantipeptide tikitericin [451].

On the opposite side of the spectrum, polar environments are also promising sources of secondary metabolite producers. *Salegentibacter* sp., isolated from the bottom of the Arctic Ocean produces twenty-five extracellular aromatic nitro containing compounds, seven of which are completely novel and some of which exhibit antimicrobial activity. Two novel rhamnolipids with high bioactivity against *Burkholderia* strains were isolated from the Antarctic isolate *Pseudomonas* sp BNT1 [452]. Mixirins A (Figure 20) B and C, on the other hand, are the product of a *Bacillus* strain isolated from the sea mud near the Arctic pole. These cyclic peptides are reported to have synergistic cytotoxic activity [453]. Arctic isolates of *Paenibacillus* sp. are the source of svalbamides A and B (Figure 20), lipodipeptides containing 3-amino-2-pyrrolidinone with potential chemopreventative activity [454].

Examples of products from halophilic bacteria include the amphiphilic siderophores loihichelins A to F (Figure 20) from *Halomonas* sp. LOB-5 [455] or the cyclic noncationic lipopeptides iturin F1, F2 and A9 from the halotolerant bacterium *Bacillus* sp. KCB14S006, which have antifungal and cytotoxic activities [456]. Deep marine sediment bacteria have also been the source of novel metabolites, like 7-O20 E-butenoyl macrolactin A, 7,13-epoxyl macrolactin A (Figure 20), (with antifungal and anti-inflammatory activities, respectively) as well as some novel ansamycins and trienomycins, all produced by *Bacillus subtilis* B5 isolated from the Pacific Ocean floor [457,458], or the indole alkaloid bacilsubteramide A (Figure 15) produced by *Bacillus subterraneus* [459]. Other environments, such as acidic mine drainage areas are also the source of novel antibiotics, and while most of the producers are reported to be fungi, there are exceptions like the case of glionitrin A (Figure 20), a diketopiperazine molecule with antimicriobial and cytotoxic activity produced by cocultures of *Aspergillus fumigatus* and *Sphingomonas* sp. Notably this compound is not produced by either of those species when cultured on their own and the actual biosynthetic origin of this molecule remains unknown [460]. All these examples, along with many other natural products coming from extremophilic and extremotolerant bacteria, highlight the potential of these niches. The application of all the techniques and tools developed while screening common mesophilic microorganisms should fast-track the discovery of further microbial and chemical diversity in these environments.

## 15. Conclusions

Throughout this review we have aimed to address the diversity of alternative sources of bacterial specialized metabolites, which hold a vast potential for the development of new drugs. Although a superficial look at the taxonomic diversity of the bacteria presented in this review as interesting secondary metabolite producers might suggest that any group of bacteria holds potential, that is not necessarily always the case. We have shown that exploration of deeply branching groups of bacteria increases the chances of finding chemical and biosynthetic novelty, such as unusual PKS and NRPS assembly line architectures, or novel RiPP families, but it is clear that the ecological context is the major driver in the evolution of diverse secondary metabolite repertoires. Complex environments in which multiple interspecific interactions occur seem to be excellent sources of talented bacteria, as demonstrated by the soil exploration in the beginning of the golden era of antibiotic discovery. The selective pressures involved in the establishment of both beneficial and antagonistic relationships between bacteria and eukaryotic organisms or other bacteria seem to positively correlate with the production of useful specialized metabolites. Similarly, bacteria showing complex life cycles or able to switch between different lifestyles are generally good candidates to be producers of interesting compounds, given the important adaptive roles that many specialized metabolites play. This often, but not always, correlates with larger genomes in which substantial sections are devoted to secondary metabolism. As ecological challenges drive the evolution of specialized metabolites, the understanding of bacterial ecology should be key to prioritize where to seek novel chemical scaffolds with unprecedented biological activities. This endeavour should be significantly boosted by the continuous development and upgrade of sequencing, bioinformatics, fermentation, metabolomics and genetic manipulation techniques that have helped us explore the still dark corners of the bacterial diversity map.

## Figures and Tables

**Figure 1 antibiotics-11-00195-f001:**
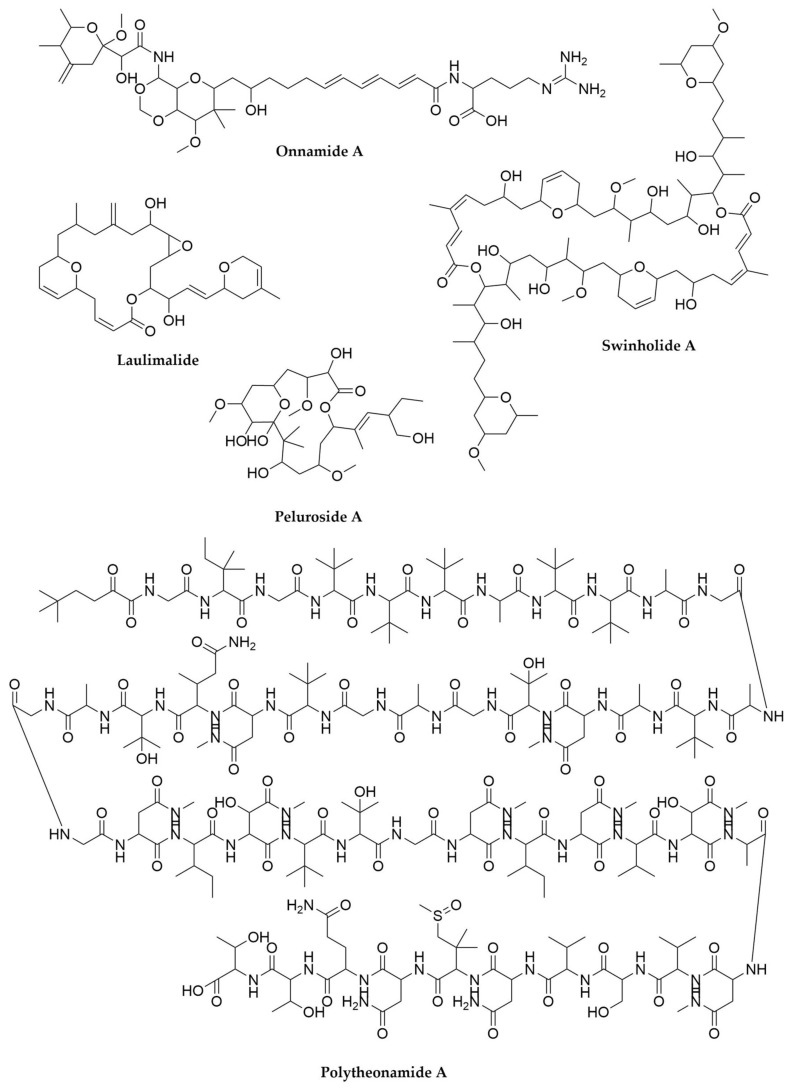
Representative compounds produced by sponge-associated bacteria.

**Figure 2 antibiotics-11-00195-f002:**
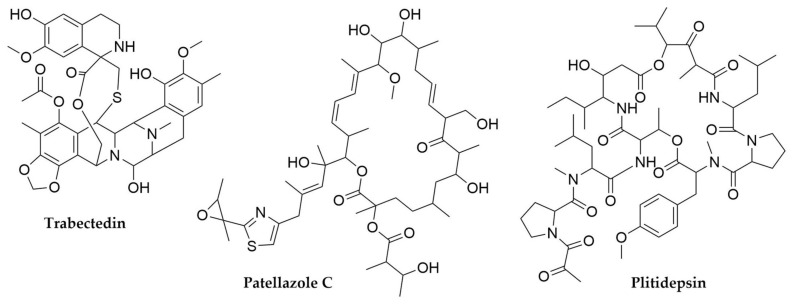
Representative compounds produced by tunicate-associated bacteria.

**Figure 3 antibiotics-11-00195-f003:**
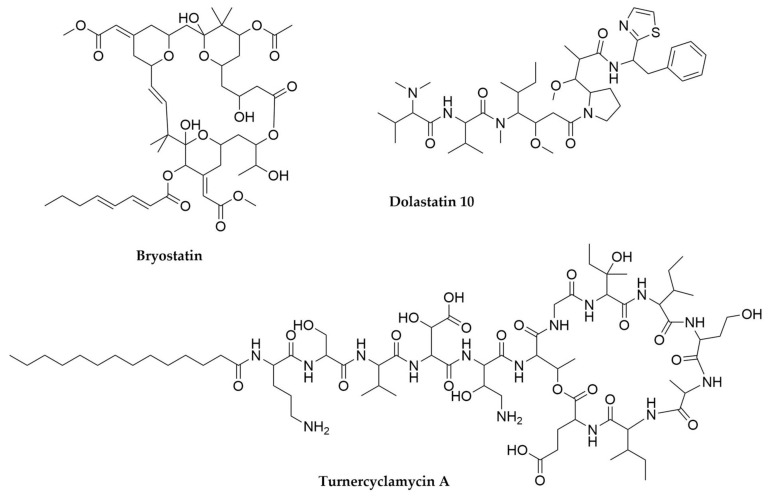
Representative compounds produced by bacteria associated to other groups of marine invertebrates.

**Figure 4 antibiotics-11-00195-f004:**
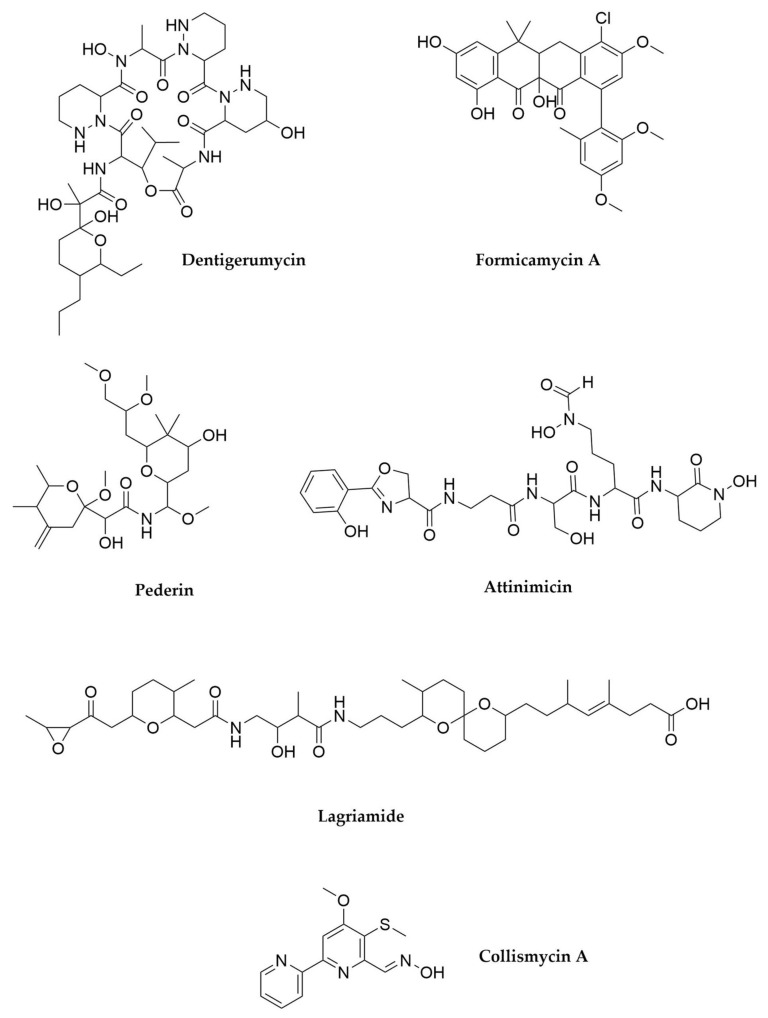
Representative compounds produced by insect-associated bacteria.

**Figure 5 antibiotics-11-00195-f005:**
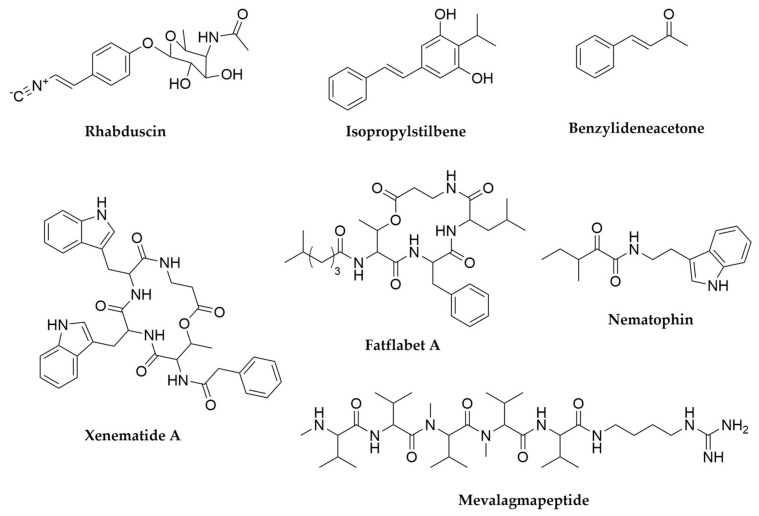
Representative compunds produced by *Xenorhabdus* and *Photorhabdus*.

**Figure 6 antibiotics-11-00195-f006:**
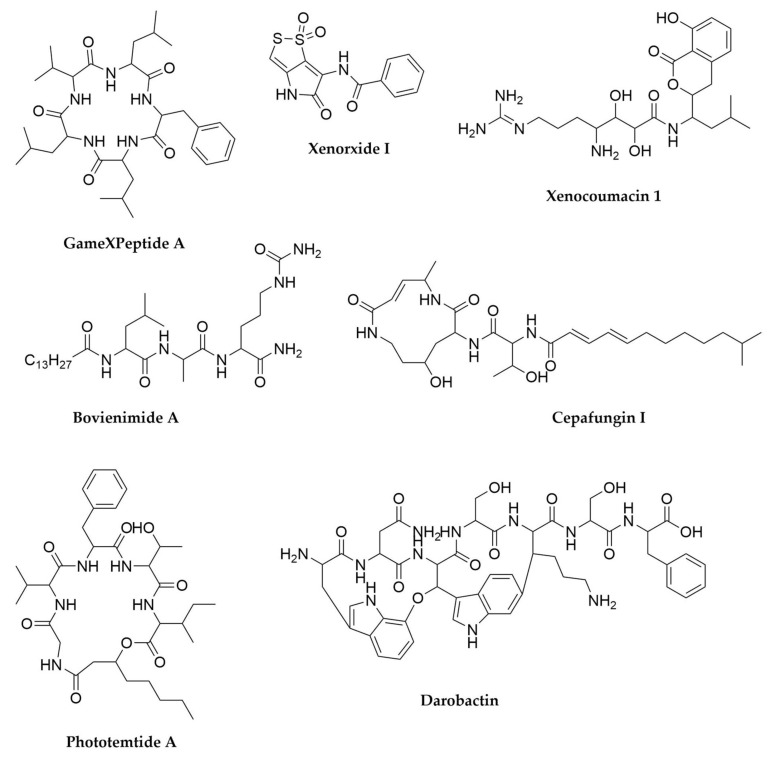
Additional representative compounds produced by *Xenorhabdus* and *Photorhabdus*.

**Figure 7 antibiotics-11-00195-f007:**
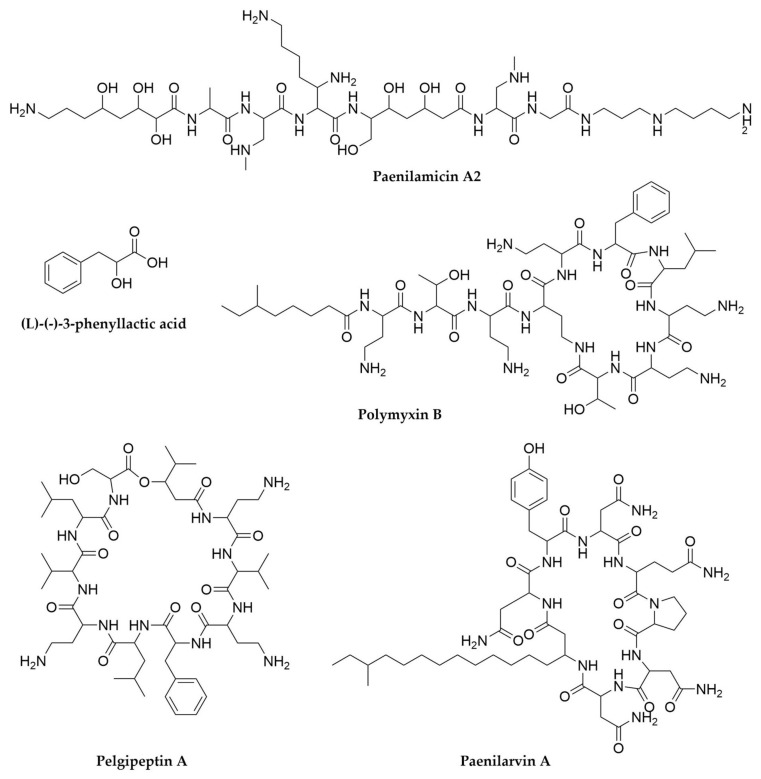
Representative compounds produced by *Paenibacillus*.

**Figure 8 antibiotics-11-00195-f008:**
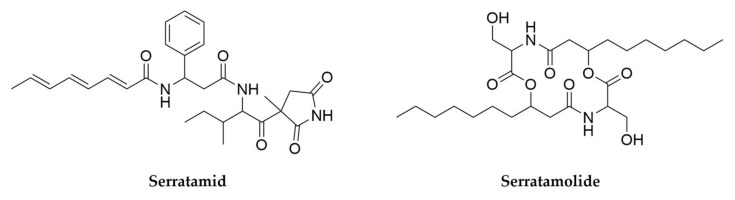
Representative compounds produced by *Serratia*.

**Figure 9 antibiotics-11-00195-f009:**
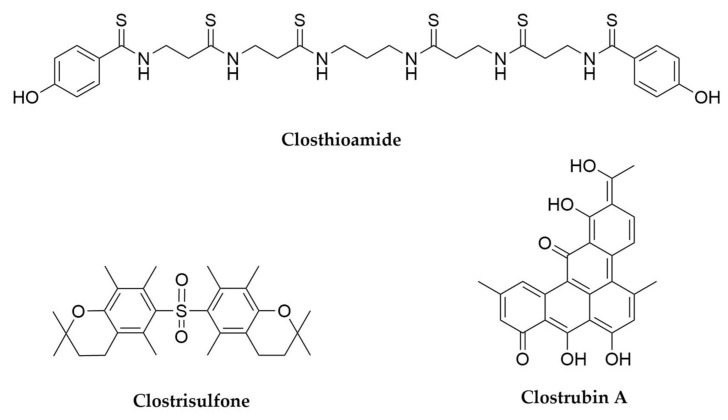
Representative compounds produced by anaerobic bacteria.

**Figure 10 antibiotics-11-00195-f010:**
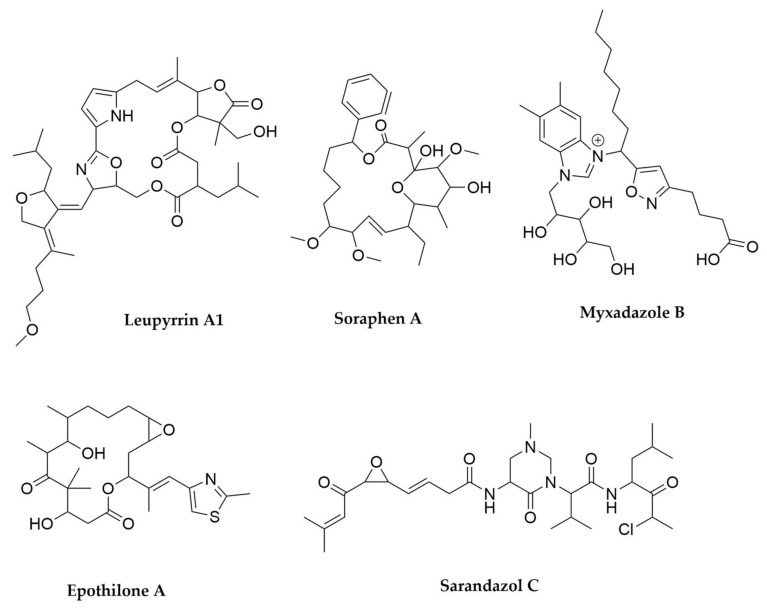
Representative compounds produced by Myxobacteria.

**Figure 11 antibiotics-11-00195-f011:**
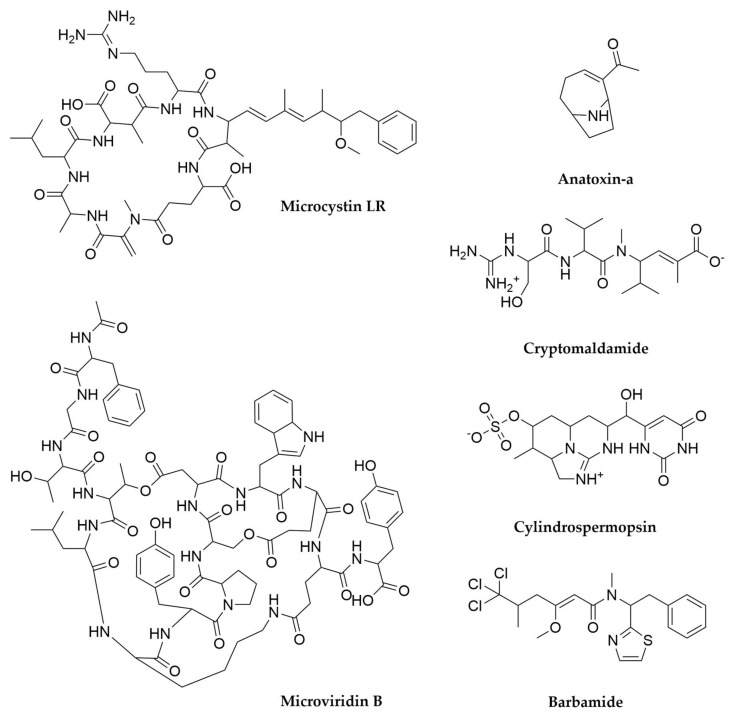
Representative compounds produced by Cyanobacteria.

**Figure 12 antibiotics-11-00195-f012:**
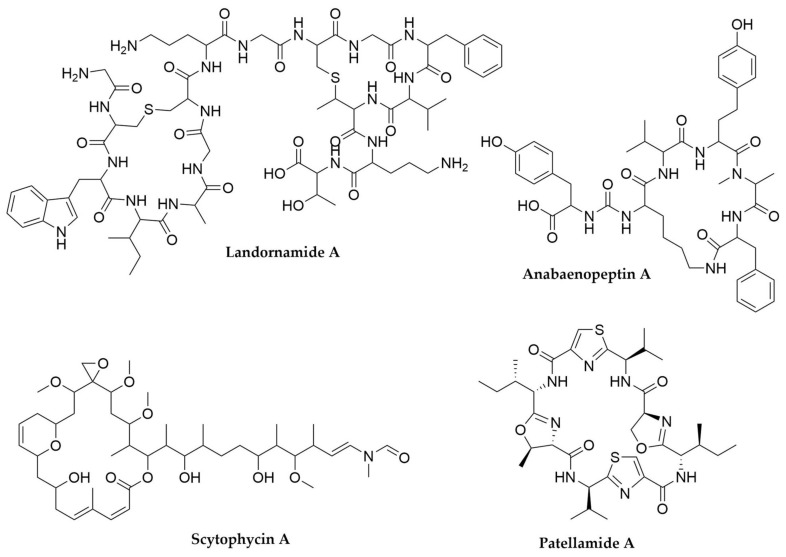
Additional representative compounds produced by Cyanobacteria.

**Figure 13 antibiotics-11-00195-f013:**
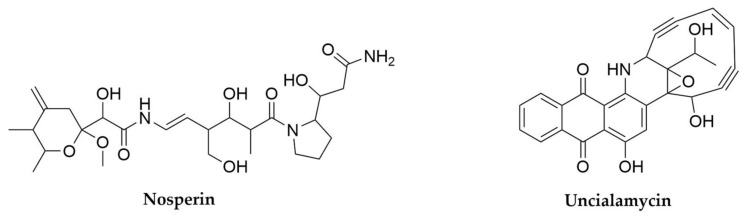
Representative compounds produced by lichen-associated bacteria.

**Figure 14 antibiotics-11-00195-f014:**
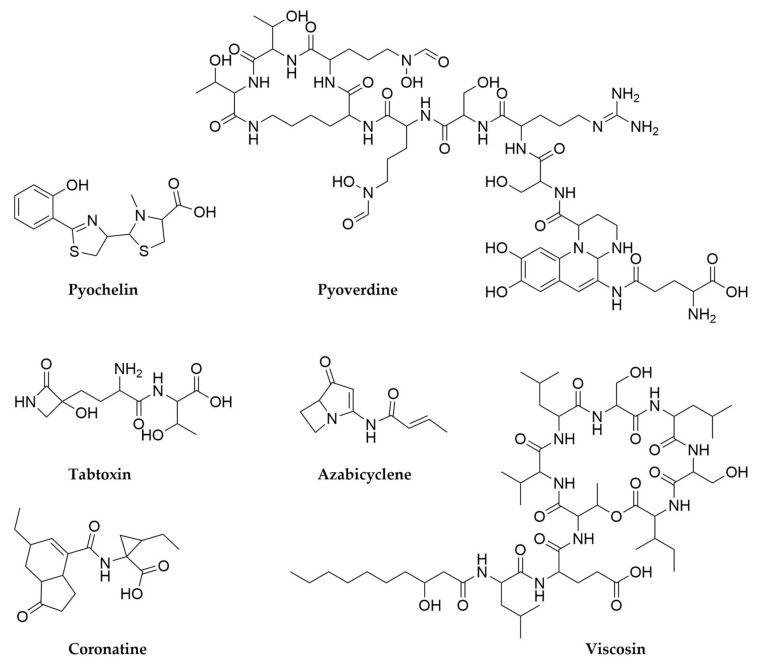
Representative compounds produced by *Pseudomonas*.

**Figure 15 antibiotics-11-00195-f015:**
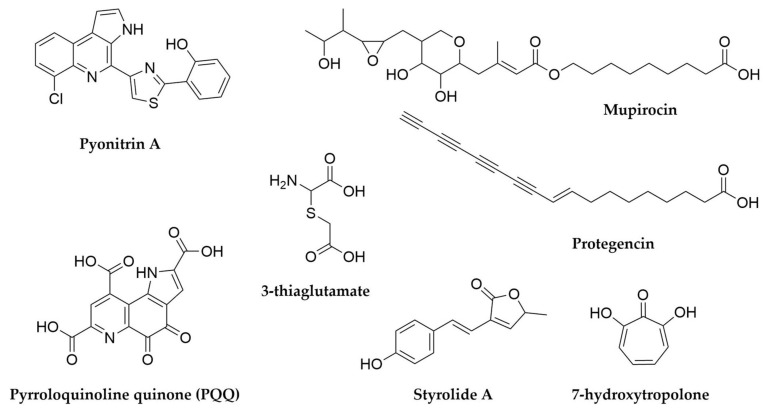
Additional representative compounds produced by *Pseudomonas*.

**Figure 16 antibiotics-11-00195-f016:**
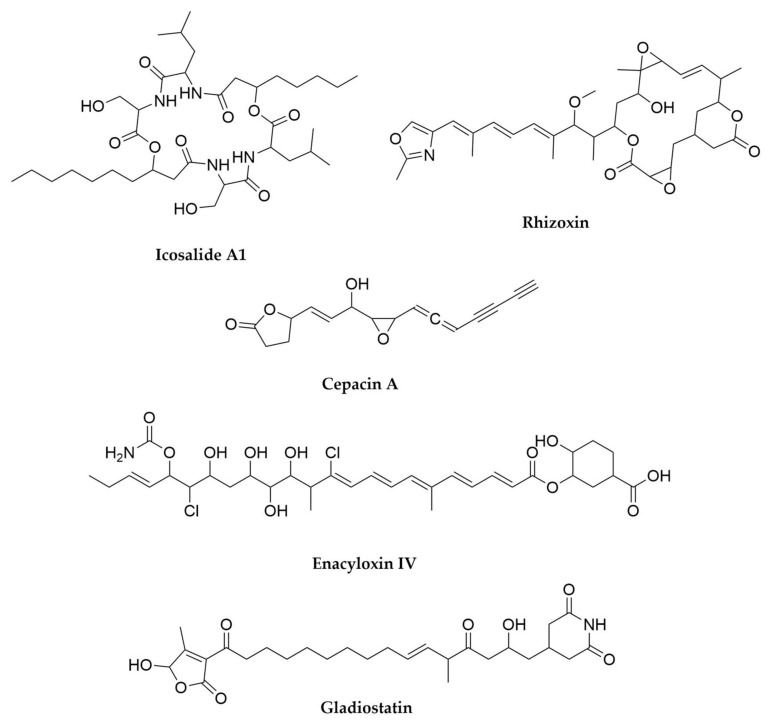
Representative molecules produced by *Burkholderia*.

**Figure 17 antibiotics-11-00195-f017:**
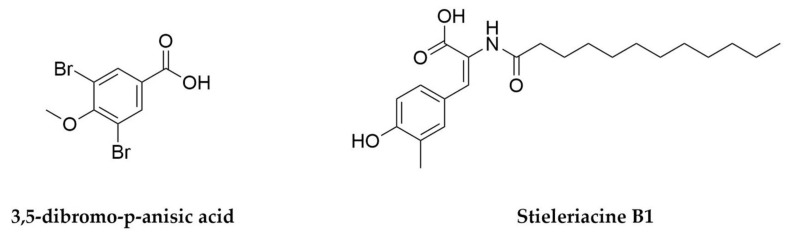
Representative molecules produced by Planctomycetes.

**Figure 18 antibiotics-11-00195-f018:**
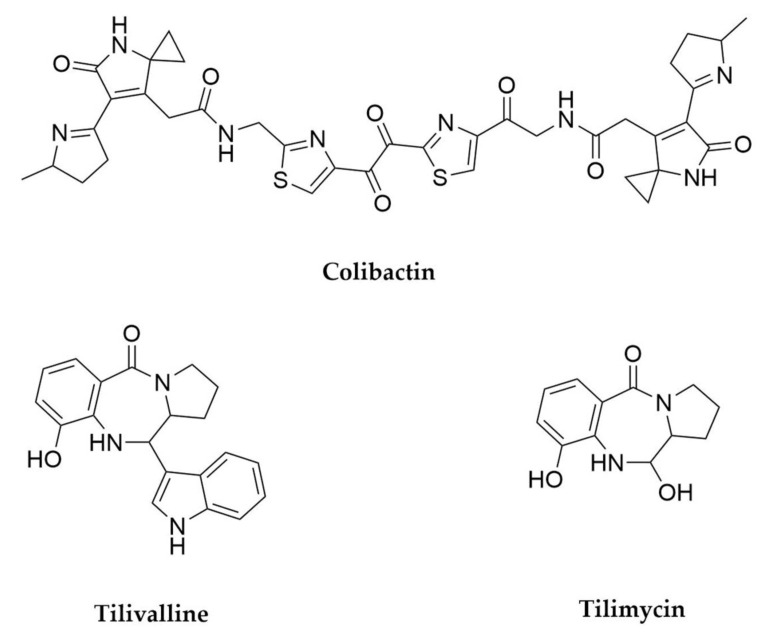
Representative compounds produced by mammalian gut bacteria.

**Figure 19 antibiotics-11-00195-f019:**
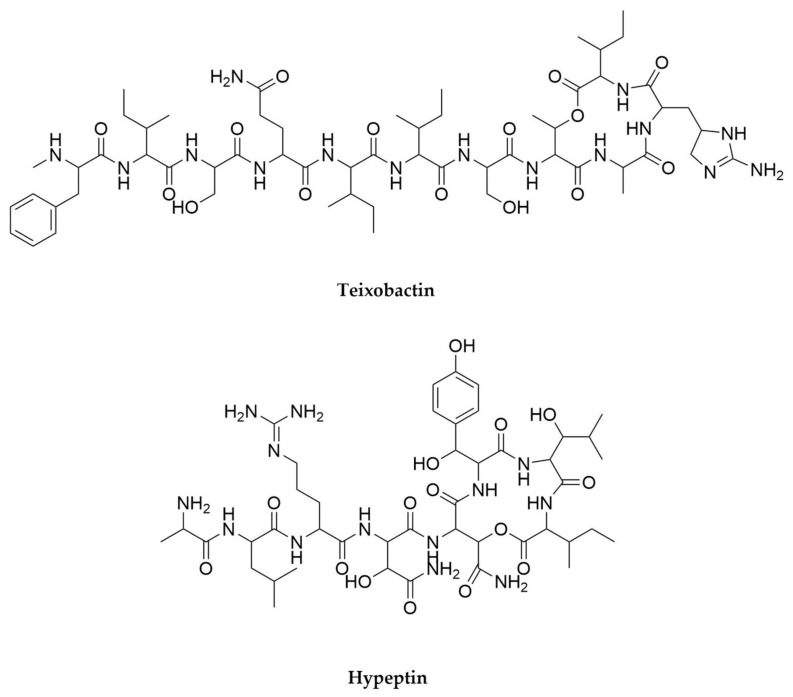
Representative compounds produced by previously unculturable bacteria from the soil cultivated using the iChip technology.

**Figure 20 antibiotics-11-00195-f020:**
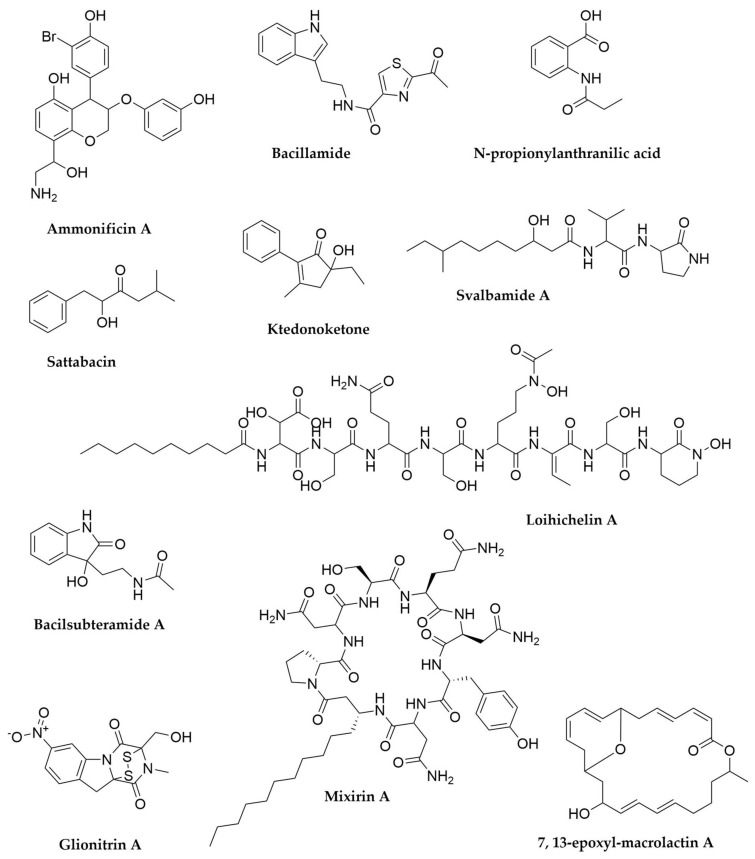
Representative compounds produced by extremophilic bacteria.

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
