# Peer review of "Beyond Soil-Dwelling Actinobacteria: Fantastic Antibiotics and Where to Find Them"

_antibiotics, 2022, doi:10.3390/antibiotics11020195_

Round 1
Reviewer 1 Report
Authors did great work, and as result - this good review. Great thanks to them, I was great pleasure and interest reading.
However, a few minor tweaks are required to make this review better.
Fig.1: The compound "peluroside A" mistaken named "peloruside A", please, fix it.
Lines 454, 458, 461, 464, 468, 470, 472, 476, 479, 483, 486, 489, 492, 494, 498, 502, 504, 506-512, 520, 523: the links font is different from the main text font.
Fig. 10: The caption to figure 10 on next page
For ease of reading and perception, I recommend dividing some figures into two, for example, Figures 4, 5, 6, 9 and 10. Perhaps this will also solve the problem of "dangling" lines.
Author Response
First, we would like to thank all the reviewers for their useful (and positive!) feedback. We were not intending to perform a comprehensive review of all the bacterial groups that are being explored as an alternative to soil-dwelling Actinobacteria, as such effort would have taken a whole book with thousands of references. Instead, our objective was to cover a general gap in literature and to provide the reader with general clues about the most important taxonomic groups and compounds, remarking the importance of ecological selective pressures that have led to the evolution of structurally unprecedented compounds with novel bioactivities. We also have tried to remark the importance of the technical novelties that have facilitated the access to previously unexplored groups of bacteria. It is possible that we got carried away with some groups more than with others, probably based in our personal work experience, but we do not think that trimming bits of the review would improve its overall message.
Reviewer 1 comments:
Authors did great work, and as result - this good review. Great thanks to them, I was great pleasure and interest reading.
Thanks a lot for such an encouraging review. We hope the changes we have introduced will make it even more enjoyable. These changes include a new lichen-associated bacteria section and a small expansion of the Myxobacteria section, as probably the importance of these microorganisms deserved a few more examples about the fascinating compounds they produce.
However, a few minor tweaks are required to make this review better.
Fig.1: The compound "peluroside A" mistaken named "peloruside A", please, fix it.
Well spotted! Thanks a lot. We have been trying to correct as much typos as possible.
Lines 454, 458, 461, 464, 468, 470, 472, 476, 479, 483, 486, 489, 492, 494, 498, 502, 504, 506-512, 520, 523: the links font is different from the main text font.
We did not find that when rechecking our file, may it be a problem with the pdf conversion? Anyway, we have tried to homogenise all the styles throughout the manuscript, including the reference list. Please let us now if there is still a problem with this.
Fig. 10: The caption to figure 10 on next page
Corrected, we have tried to fit all the figures as good as possible.
For ease of reading and perception, I recommend dividing some figures into two, for example, Figures 4, 5, 6, 9 and 10. Perhaps this will also solve the problem of "dangling" lines.
Thanks, that is a good suggestion. We have split into two figures 5, 6, 9 and 10. We think figure 4 does not have enough molecules to make it uncomfortable for the reader. We had our doubts about figure 1, due to the huge size of the polytheonamide molecule, but we have decided to keep its original format due to the moderate number of molecules in the figure. We hope after the professional edition things will look good, if the manuscript is finally accepted for publication.
Reviewer 2 Report
Very attractive title and comprehensive summary, I admire the author's great effort to organize such a meaningful work. Of course, I should be critical of the manuscript in order to improve its quality.
Main conmments:
- As an emerging source of secondary metabolites in recent years, lichen symbiotic bacteria should also be summarized as a group, like Insect-associated Bacteria.
- The compound structure and text on all pictures in the manuscript are false. Although this is a technical problem, please pay attention.
- The manuscript summarizes 12 types of bacterial secondary metabolite sources other than soil sources, but not every type is comprehensively summarized. Please check carefully. For the grouping of "Extremophilic bacteria", please cite the following article: Antimicrobial activity and biosynthetic potential of cultivable actinomycetes associated with Lichen symbiosis from Qinghai-Tibet Plateau. 10.1016/j.micres.2020.126652.
Minor shortcomings:
- The literature citation marks from lines 454 to 523 are not standardized, please check.
- Some of the journal names in the references are sometimes full, sometimes abbreviated, please unify them.
Author Response
First, we would like to thank all the reviewers for their useful (and positive!) feedback. We were not intending to perform a comprehensive review of all the bacterial groups that are being explored as an alternative to soil-dwelling Actinobacteria, as such effort would have taken a whole book with thousands of references. Instead, our objective was to cover a general gap in literature and provide the reader with general clues about the most important taxonomic groups and compounds, remarking the importance of ecological selective pressures that have led to the evolution of structurally unprecedented compounds with novel bioactivities. We also have tried to remark the importance of the technical novelties that have facilitated the access to previously unexplored groups of bacteria. It is possible that we got carried away with some groups more than with others, probably based in our personal work experience, but we do not think that trimming bits of the review would improve its overall message.
Reviewer 2 comments
Very attractive title and comprehensive summary, I admire the author's great effort to organize such a meaningful work. Of course, I should be critical of the manuscript in order to improve its quality.
Response: Thanks a lot. It took a good amount of effort indeed, but we think that after introducing the changes suggested by the reviewers, the manuscript will be even more appealing to the readers.
Main conmments:
- As an emerging source of secondary metabolites in recent years, lichen symbiotic bacteria should also be summarized as a group, like Insect-associated Bacteria.
Response: We agree that we have overlooked lichen-associated bacteria, and now we have included a section to comment on them. Thank you for pointing it out.
- The compound structure and text on all pictures in the manuscript are false. Although this is a technical problem, please pay attention.
Response: we are nor completely sure what the reviewer means with “false”. We did not include stereochemistry in the figures as its determination was not available for many compounds, so we decided to keep plain structures. We have changed the names of the compounds to palatino linotype to match their style with the rest of the text. We hope that is all right, but please let us know if any problem persists.
- The manuscript summarizes 12 types of bacterial secondary metabolite sources other than soil sources, but not every type is comprehensively summarized. Please check carefully. For the grouping of "Extremophilic bacteria", please cite the following article: Antimicrobial activity and biosynthetic potential of cultivable actinomycetes associated with Lichen symbiosis from Qinghai-Tibet Plateau. 10.1016/j.micres.2020.126652.
Response: we consider it would take a whole book (and a thick one) to summarize all the knowledge about alternative bacterial sources of natural products. Our intention was to guide the reader through the taxonomy and ecology of particularly interesting groups, remarking new techniques and the most import compounds. As we said in the general response to the reviewers, we may have got carried away too much with a couple of groups, due to our personal interest on them We have expanded the myxobacteria section a little with a few more examples, as probably the importance of this particular group of bacteria deserved a few more comments. We have included the requested reference, but including it in the new lichens section instead than among the extremophiles.
Minor shortcomings:
- The literature citation marks from lines 454 to 523 are not standardized, please check.
Response: Reviewer 1 also pointed this out, but we have not found the problem. We suspect it could be related to the pdf conversion, but we have made an effort to homogenise the style throughout the whole manuscript. Please let us now if the problem persists.
- Some of the journal names in the references are sometimes full, sometimes abbreviated, please unify them.
Response: Well spotted. For some reason our reference manager did strange stuff with the names of a couple of journals (mainly PNAS and ACIE). We have corrected them manually, as well as other small reference manager issues. We hope everything is fine now.
Reviewer 3 Report
Title: The authors did justice with the title. They discussed many types of bacteria as a source of bioactive compounds or secondary metabolites beyond soil-dwelling actinobacteria.
The manuscript may shorten by reducing the details discussion of chemical classes rather more specific on sources, bioactivity and compound types. I didn’t find any table or graph which could be very interesting for this manuscript in order to get all the information together.
In this review, the importance of exploring bacteria from various sources came out nicely as a resource of getting novel bioactive compounds as future antibiotics and drugs.
Minor: line 7: rather unvaluable it’s better to write invaluable
Line 77 to 81 require reference
Line 88-90 require reference
If possible avoid too many references. And put reference where require.
References need to be consistent in a specific format particularly please make changes all the journal name in similar style.
Author Response
First, we would like to thank all the reviewers for their useful (and positive!) feedback. We were not intending to perform a comprehensive review of all the bacterial groups that are being explored as an alternative to soil-dwelling Actinobacteria, as such effort would have taken a whole book with thousands of references. Instead, our objective was to cover a general gap in literature and provide the reader with general clues about the most important taxonomic groups and compounds, remarking the importance of ecological selective pressures that have led to the evolution of structurally unprecedented compounds with novel bioactivities. We also have tried to remark the importance of the technical novelties that have facilitated the access to previously unexplored groups of bacteria. It is possible that we got carried away with some groups more than with others, probably based in our personal work experience, but we do not think that trimming bits of the review would improve its overall message.
Reviewer 3 comments:
Title: The authors did justice with the title. They discussed many types of bacteria as a source of bioactive compounds or secondary metabolites beyond soil-dwelling actinobacteria.
Response: Thanks for your positive comments. We hope we have been able to improve the manuscript thanks to the comments of all the reviewers. These changes include a new lichen-associated bacteria section and a small expansion of the Myxobacteria section, as probably the importance of these microorganisms deserved a few more examples about the fascinating compounds they produce.
The manuscript may shorten by reducing the details discussion of chemical classes rather more specific on sources, bioactivity and compound types. I didn’t find any table or graph which could be very interesting for this manuscript in order to get all the information together.
We consider that commenting on chemical classes would be useful to refresh the knowledge of a general reader. Generally, the table would be a good idea if the focus of the review was slightly different. However, we are not intending to perform a comprehensive review of all the groups but to provide the reader with general guidelines and orientate them into a very complex topic.
In this review, the importance of exploring bacteria from various sources came out nicely as a resource of getting novel bioactive compounds as future antibiotics and drugs.
Thanks again for such appositive comment, we appreciate it.
Minor: line 7: rather unvaluable it’s better to write invaluable
Corrected. As non-native English speakers we were not aware of the slightly different meaning of these two words. Thanks a lot for the comment.
Line 77 to 81 require reference
Response: we have added one reference to support those claims.
Line 88-90 require reference
Response: we have added two references to support those claims
If possible, avoid too many references. And put reference where require.
Response: we think that given the wide topic the number of references is not excessive. Maybe we got carried away in a couple of sections, but that means just more detail for the interested reader. In general, we would say the reference marks are at the right places, but sometimes that of course could be a matter of debate.
References need to be consistent in a specific format particularly please make changes all the journal name in similar style.
Well spotted. We have now manually corrected a few strange mistakes and inconsistencies introduced by the reference manager in the reference list. We hope now everything is OK.